# Leveraging Physics-Based Models for Rapid Adaptation in Reinforcement Learning

## Abstract

A central challenge in reinforcement learning (RL) is achieving agents that generalize and adapt to new tasks and conditions. Many works address this via offline RL which is constrained by dataset coverage, or online RL which requires costly and potentially unsafe exploration. We propose a framework for rapid adaptation of RL agents by augmenting model-based RL with physics-informed data augmentation. Specifically, we use lightweight analytical models to generate stable, physics-grounded rollouts that complement real interaction data and allows the model-based RL agent to adapt in just a few trials. We validate our approach in autonomous racing, an extreme testbed with fast dynamics and strict safety constraints, using Assetto Corsa paired with lightweight vehicle models for data augmentation. Across diverse tracks and surfaces, our method achieves faster convergence, lower lap times, and fewer incidents than a set of strong baselines. Although demonstrated in racing, our framework is domain-agnostic, offering a practical path to data-efficient control wherever simple models exist as priors.

## 1 Introduction

Reinforcement learning (RL) offers a principled framework for data-driven optimization of control policies in unknown environments, and has achieved remarkable results when coupled with large quantities of interaction data (Berner et al., 2019; Schrittwieser et al., 2020; Wurman et al., 2022). For example, Wurman et al. (2022) demonstrates that an RL policy outraces the world's best human drivers in Gran Turismo after $45,000$ hours of online interaction. However, the poor data-efficiency and general brittleness of existing RL algorithms greatly limits their applicability to domains where *(i)* data collection is costly (*e.g.* robotics —or driving in the real world), *(ii)* ensuring safe exploration during training is important, or *(iii)* properties of the task frequently change (*e.g.* a drift in dynamics).

As a result, RL practitioners often face a tradeoff between dataset diversity and interaction budget. Offline RL is attractive due to its reliance on existing data but may lead to (unsafe) extrapolation errors when the policy encounters out-of-distribution states (Kumar et al., 2020; Kostrikov et al., 2021); such extrapolation errors can be corrected with additional data collected by online RL (Nair et al., 2021; Feng et al., 2023; Ball et al., 2023) but that too can be unsafe. In domains such as high-speed racing where mistakes can be costly, this tradeoff often produces one of two undesirable outcomes: systems that require thousands of laps to achieve stable performance which is unrealistic for real-world racing, or controllers that rely on highly accurate physics models which limits flexibility and may potentially lead to suboptimal policies.

The challenge is to prepare policies that leverage existing datasets to transfer to unseen tasks and adapt rapidly under strict interaction budgets.

Dyna-Q (Sutton, 1990; Janner et al., 2019; Yu et al., 2020), have proposed using synthetic rollouts from learned dynamics and have shown that model-based rollouts can significantly improve sample efficiency. However, they are prone to compounding error, so rollouts are typically kept short to remain reliable. For control and model-based RL (MBRL), short rollouts are typically preferred to avoid compounding errors and because frequent feedback is available. In simulation, offered by platforms such as MuJoCo (Todorov et al., 2012) and NVIDIA Isaac Gym (Makoviychuk et al., 2021), analytical models are used because they remain stable over long horizons, whereas learned models often accumulate error and drift. We take this perspective and argue that data augmentation

*Figure 1.* **Assetto Corsa.** We evaluate in autonomous racing as an extreme setting. Assetto Corsa, a highly realistic racing simulator, serves as a controlled proxy for real-world racing.

can be more effective when viewed as a simulation problem rather than a control problem: analytical models provide reliable long-horizon rollouts.

We therefore propose physics-informed augmentation using lightweight analytical models. These models are cheap, stable, and physics-grounded, producing long-horizon synthetic rollouts that align with high-fidelity simulators. By merging them into replay buffers and combining with MBRL, we create a natural bridge: synthetic diversity from physics rollouts, efficient adaptation from MBRL. Importantly, this approach enables augmentation in situations never encountered by the agent.

We evaluate in autonomous racing as an extreme setting. The Assetto Corsa racing simulator (AC-Gym) (Remonda et al., 2024), shown in Figure 1, provides realism, while an ODE (Ordinary differential equation) vehicle model cheaply expand data. We treat sim-to-sim transfer (ODE→Assetto Corsa) as a controlled proxy for sim-to-real, isolating algorithmic choices without hardware confounds. Unlike prior work requiring massive interactions (Wurman et al., 2022), our method achieves safe adaptation in a few trials. While racing is the testbed, the framework is domain-agnostic and applicable wherever lightweight models exist (robotics, locomotion). *We are committed to open-sourcing our code upon acceptance.* **Our contributions can be summarized as follows:**

- **Physics-informed data augmentation MBRL**: A framework where analytical rollouts expand replay buffers and MBRL integrates them with simulator data, yielding better generalization, adaptation, and sample efficiency.

- **Comprehensive evaluation**: We evaluate the framework in the demanding ACGym racing benchmark, studying how track diversity and surface conditions impact adaptation and highlighting the key components for success under limited interaction.

- **System improvements**: We extend the high-fidelity racing simulator ACGym with several engineering enhancements.

- **Reproducible benchmark**: We release source code and datasets covering more than 20 tracks under multiple conditions, totaling >1000 hours of AC driving data and a comparable amount of synthetic ODE rollouts, enabling future research on few-shot and sim2sim-to-real adaptation.

Our experiments show that physics-informed MBRL improves adaptation and reduces failures after only a few online episodes. Overall, physics-informed MBRL with lightweight model augmentation is a practical approach for high-performance control in hazardous scenarios and provides a benchmark for generalization in high-speed racing.

## 2 PRELIMINARIES

**Problem Definition.** We formulate autonomous racing as a Partially Observable Markov Decision Process (POMDP) (Bellman, 1957; Kaelbling et al., 1998) $\langle \mathcal{S}, \mathcal{A}, \mathcal{T}, r, \gamma \rangle$, where $\mathcal{S}$ denotes the state space, $\mathcal{A}$ is the action space, $\mathcal{T}: \mathcal{S} \times \mathcal{A} \mapsto \mathcal{S}$ is the (unknown) transition model, $r: \mathcal{S} \times \mathcal{A} \mapsto \mathbb{R}$ is a reward function, and $\gamma \in [0, 1)$ is the discount factor. We approximate states as a sequence of the last $k$ observations received from the environment. A RL policy $\pi$ is trained to select actions $\mathbf{a}_t \sim \pi(\cdot|\mathbf{s}_t)$ at each time step $t$ such that the expected sum of discounted rewards $\mathbb{E}_\pi \left[ \sum_{t=0}^\infty \gamma^t r(\mathbf{s}_t, \mathbf{a}_t) \right]$ is maximized.

**Model-Based RL.** In MBRL, the agent learns a predictive model of environment dynamics and leverages it for planning. This improves sample efficiency over model-free methods and enables transfer by reusing learned dynamics. We adopt TD-MPC2 (Hansen et al., 2023) as our backbone.

TD-MPC2 learns a latent world model from sequential data and performs short-horizon planning with Model Predictive Control. Its end-to-end training and efficient latent-space planning make it well-suited for multitask learning and diverse datasets.

**Offline RL and Adaptation.** Given a static dataset $\mathcal{D}_{\text{offline}}$, offline RL seeks to learn $\pi$ without additional interaction. (Levine et al., 2020) The main challenge is out-of-distribution generalization when $\pi$ reaches unseen states. A common strategy to mitigate this issue is offline pretraining followed by online adaptation, where a world model is initialized on $\mathcal{D}_{\text{offline}}$ and then refined with limited new interaction $\mathcal{D}_{\text{online}}$ (Feng et al., 2023).

**ODE Model.** We employ an analytical vehicle model described by ordinary differential equations of the form $\dot{\mathbf{x}} = g(\mathbf{x}, \mathbf{u}; \phi)$, where $\mathbf{x}$ is the vehicle state, $\mathbf{u}$ the control inputs, and $\phi$ the physical parameters (mass, inertia, etc). The model extends the bicycle formulation with aerodynamics, drivetrain dynamics, and a Pacejka tire force models. Trajectories are simulated via fourth-order Runge–Kutta integration, enabling stable synthetic rollouts for augmentation. (Raji et al., 2023)

**Assetto Corsa Gym.** For high-fidelity simulation, we use ACGym (Remonda et al., 2024), an open-source framework built on the Assetto Corsa racing simulator. It provides realistic vehicle dynamics and track geometries across several cars and laser-scanned tracks, and exposes a `gym`-compliant API. The action space is continuous (steering, throttle, brake), and the observations include proprioceptive signals and environmental features.

# 3 PHYSICS-INFORMED DATA AUGMENTATION WITH MODEL-BASED RL

We propose a general framework that combines physics-informed augmentation with lightweight analytical models and state-of-the-art MBRL. Our design builds directly upon two lines of work: **FOWM** (Feng et al., 2023) demonstrates that pretraining a world model on offline data and then fine-tuning with limited online interaction yields efficient adaptation. Key to its success is planner regularization to mitigate extrapolation error under distribution shift. **TD-MPC2** (Hansen et al., 2023) is a MBRL method that performs local trajectory optimization in the latent space of a decoder-free world model, achieving strong performance across a diverse set RL tasks with a single set of hyperparameters. It scales reliably with data and model size, and supports multi-task learning without per-task tuning. We combine these insights: FOWM motivates using an offline-to-online training paradigm, while TD-MPC2 provides a robust RL backbone for bridging heterogeneous datasets (ODE + high-fidelity simulator) and adapting to new tasks under a strict interaction budget.

## 3.1 PHYSICS-INFORMED DATA AUGMENTATION

Offline RL is fundamentally constrained by dataset diversity, and synthetic rollouts from learned dynamics are only reliable at short horizons. We instead use analytical ODE models: closed-form vehicle equations that are interpretable, cheap to simulate, and stable over long rollouts. These properties make them ideal for data augmentation: ODE trajectories can be generated at scale, remain consistent with physical priors, and cover situations rarely encountered in demonstrations or high-fidelity simulation. By merging them into the offline buffer, the agent gains access to a much broader range of state-action transitions while preserving consistency with real dynamics.

## 3.2 MODEL-BASED RL AS THE BRIDGE

MBRL naturally unifies heterogeneous datasets by learning latent dynamics and planning across trajectories, unlike model-free RL which often requires per-task tuning. We use TD-MPC2 as backbone for its scalability and robustness: it trains an implicit latent world model, plans in latent space, scales to large multi-task datasets with a single set of hyperparameters, and improves reliably with more data and model capacity. In our framework, the offline stage pretrains on combined ODE + simulator buffers, while the online stage collects a few trajectories in the target environment to close the sim2sim gap and adapt efficiently.

## 3.3 APPLICATION TO RACING

Autonomous racing is a natural testbed for our framework: it combines high-speed dynamics, strict safety constraints, and frequent domain shifts across tracks and surfaces. Unlike manipulation or

locomotion, where online interaction is relatively safe, racing demands rapid adaptation under limited interaction budgets making the perfect setting to evaluate our proposed method. We use Assetto Corsa as a controlled proxy for real, isolating algorithmic contributions without hardware confounds.

We combine the high-fidelity simulator Assetto Corsa for realism with a lightweight ODE bicycle model (aerodynamics, drivetrain, Pacejka tires) for cheap, large-scale rollout generation. Track geometry (borders, curvature, racing line) is inherited from AC, while dynamics come from the ODE model, ensuring compatibility between synthetic and high-fidelity data.

In practice, we replace only the dynamics step of the environment with an ODE model, while geometry channels (distance to borders, curvature look-ahead, distance to the racing line) are inherited directly from ACGym given the dynamics state. This modular design allows any dynamics model to be swapped in by redefining the step function, while the observation interface remains unchanged.

Synthetic trajectories are collected by running SAC in the ODE environment, reusing ACGym's logging infrastructure so that synthetic data share the same format as high-fidelity rollouts. These datasets are then merged with Assetto Corsa trajectories into a single offline buffer for pretraining.

---

**Algorithm 1** PIA-MBRL: Physics-Informed Augmentation with MBRL

1: **Inputs:** $\mathcal{D}_{\text{offline}}$ (datasets), environment $E$, offline steps $N_{\text{pre}}$, episodes $N_{\text{ep}}$, batch size $b$
2: **Buffers:** $B_{\text{offline}} \leftarrow \emptyset, \; B_{\text{online}} \leftarrow \emptyset$
3: **(A) Build offline buffer**
4: **for** each dataset $D \in \mathcal{D}_{\text{offline}}$ **do**
5:     **for** each transition $(o, a, r, o') \in D$ **do**
6:         push $(o, a, r, o')$ into $B_{\text{offline}}$
7: **(B) Offline pretraining**
8: **for** $t = 1$ to $N_{\text{pre}}$ **do**
9:     $\mathcal{B} \leftarrow \text{sample}(B_{\text{offline}}, b)$
10:     update_model($\mathcal{B}$)
11: save_checkpoint()
12: **(C) Online fine-tuning in** $E$
13: **for** episode $= 1$ to $N_{\text{ep}}$ **do**
14:     $o \leftarrow \text{reset}(E)$
15:     **while** episode not done **do**
16:         $a \leftarrow \text{select\_action}(o)$
17:         $(o', r) \leftarrow \text{step}(E, a)$
18:         push $(o, a, r, o')$ into $B_{\text{online}}$
19:         $o \leftarrow o'$
20:         */* Train-after-collect with 50/50 mixing */*
21:         $\mathcal{B}_{\text{off}} \leftarrow \text{sample}(B_{\text{offline}}, \lfloor b/2 \rfloor)$
22:         $\mathcal{B}_{\text{on}} \leftarrow \text{sample}(B_{\text{online}}, \lceil b/2 \rceil)$
23:         $\mathcal{B} \leftarrow \mathcal{B}_{\text{off}} \cup \mathcal{B}_{\text{on}}$
24:         update_model($\mathcal{B}$)

---

Full ODE model details and environment specifications are provided in Appendix C.

### 3.4 ALGORITHMIC CHOICES

We adopt several design choices which we show are necessary for stability and real-time feasibility:

**Balanced offline/online buffer mixing** For stability and faster adaptation When learning from both offline and online interaction data, we maintain separate buffers for each data source and sample such that each mini-batch contains 50% data from each source. (Feng et al., 2023)

**Planning regularization** We regularize the planner by penalizing predicted return weighted by model uncertainty, reducing extrapolation errors during finetuning. (Feng et al., 2023)

**Train-after-collect** We decouple data collection and updates, since online updates introduced control delays that harmed performance.

**Task conditioning** We encode both the track ID and the data source ID (ODE vs AC) as one-hot vectors and append them to the model input, providing explicit task conditioning across tracks and data sources. We show that this improves performance when mixing ODE and AC data.

### 3.5 SYSTEM ENGINEERING

**System.** Racing has stricter real-time restrictions than manipulation (as in FOWM). We therefore adopt a JAX implementation of TD-MPC2 (see Appendix G.4), which is compute-efficient and real-time feasible. In contrast, we find that the official PyTorch version is too slow for real-time interaction when larger models are used.

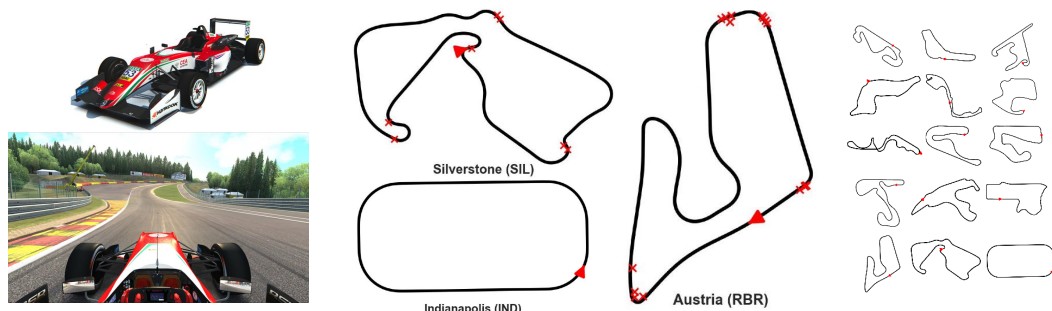

*Figure 2. Left*: F317 car and Assetto Corsa; *Middle*: test tracks; *Right*: all tracks considered.

**ACGym.** We extend ACGym in several ways, including Linux support which is crucial to deployment of Jax-based algorithms. Additionally, we rework the framework to support swapping of different dynamics. See Appendix A for more details.

### 3.6 ALGORITHM

Our method, **PIA-MBRL**, proceeds in three stages. First, we build an offline buffer from augmented datasets. Second, the model is pretrained on this buffer to acquire structured priors. Finally, the agent is fine-tuned online in the target environment, using a balanced mix of offline and online data for updates (see Algorithm 1). We follow the standard *offline pretraining + online fine-tuning* structure. Offline datasets may include high-fidelity rollouts, or ODE-generated trajectories, all stored in a buffer $B_{\text{offline}}$. The agent is pretrained by sampling minibatches from $B_{\text{offline}}$. During online interaction, new trajectories are stored in a buffer $B_{\text{online}}$. After each episode, updates use minibatches mixed 50/50 from both buffers, which stabilizes training and accelerates adaptation.

For ODE rollouts, we train a SAC agent directly in the ODE environment to generate trajectories, which are stored in the same format as AC datasets and merged into $B_{\text{offline}}$.

## 4 EXPERIMENTS

Our general method can be instantiated in any domain when a lightweight model is available. We demonstrate this in autonomous racing, using Assetto Corsa as the high-fidelity simulator and an ODE vehicle model as the lightweight model. We design experiments to test hypotheses about performance, scaling and robustness. *We are committed to open-sourcing our code upon acceptance.* **We address the following research questions:**

• **RQ1 – Effectiveness:** Does our proposed framework improve performance and safety compared to strong baselines across unseen tasks?
• **RQ2 – Robustness:** Does augmentation enhance adaptation under domain shifts, such as changes in surface grip, without explicitly modeling these variations?
• **RQ3 – Data scaling:** How does the size and diversity of pretraining datasets affect generalization to new tracks? Do larger and more varied offline buffers yield better adaptation?
• **RQ4 – Synthetic viability:** Can policies pretrained purely on synthetic ODE rollouts achieve competitive adaptation?
• **RQ5 – Mechanisms:** Which components of our framework (geometry vs. dynamics signals, planner regularization, task/context conditioning) are critical to its success?

### 4.1 TRAINING AND EVALUATION

**Tasks** We evaluate our method in Assetto Corsa Gym (ACGym) across a broad set of racing tasks and conditions. The benchmark consists of fifteen training tracks and three held-out test tracks, organized into progressively larger subsets to study data scaling. The subsets are constructed as follows: 15 tracks, grouped as: 1T = {BRN}, 3T = 1T + {MNZ, CHN}, 6T = 3T + {MON, IML, LAS}, 9T = 6T + {SUK, TSU, RAM}, 12T = 9T + {ZAN, SPA, NED}. Test = {IND, RBR, SIL}. To study robustness, we evaluate on three surface conditions: *optimal*, *green*, *dusty*. Optimal:

Perfectly track with maximum grip always. Green: Fresh, clean track that progressively gains grip with running. Dusty: Very slippery - low grip. (See Appendix B.1 and B.2 for track and surface details.) Each episode lasts up to 15k environment steps (600 seconds real time at 25Hz). Episodes terminate early if the car stalls or leaves the track.

**Metrics** Performance is measured using several metrics. We report best lap time and peak episode reward as indicators of final driving quality. To assess learning efficiency, we track the number of episodes required to achieve the first successful completion of an episode. Safety and stability are captured by number of incidents, defined as off-track excursions or crashes, and by miles per incident, which measures how far the agent drove between failures.

**Datasets** We construct two sources of pretraining data. **AC (high-fidelity).** We train SAC from scratch for 6M steps per track on ACGym to generate expert-level trajectories. These datasets cover all training splits (1T–12T). **ODE (synthetic).** We introduce a lightweight ODE environment that shares the same observation and action interface as ACGym (see Appendix C for details). This ensures compatibility with any OpenAI Gym–based algorithm. To generate data, we run the same SAC agent used for AC, but replace the physics step with the ODE model while keeping geometry and localization signals. This setup enables rapid collection of large synthetic datasets that remain fully aligned with the AC channels required for training.

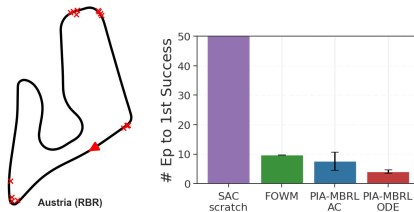

*(a)* Failures     *(b)* Eps. to first success

*Figure 3.* Left: failure locations along the track. Right: adaptation efficiency measured as episodes to first success.

**Methods TD-MPC2-scratch** trains TD-MPC2 entirely online without offline data. **PIL-MBRL-$X$** refers to our framework (Section 3), where $X$ specifies the pretraining dataset: $T$ denotes AC tracks (e.g., PIL-MBRL-6T) and *ODE* denotes ODE datasets (e.g., PIL-MBRL-3ODE). **SAC-scratch** is a model-free baseline trained online. **IQL** is an offline RL baseline trained on AC datasets. Finally, **FOWM** is an offline-to-online finetuning baseline using AC datasets.

## 4.2 RESULTS

**Physics-Informed Data Augmentation** We evaluate whether physics-informed data augmentation improves generalization across unseen tasks: IND (oval), RBR (mid-difficulty), SIL (complex), and two grip-shift variants of SIL (dusty and green). We evaluate two variants: PIA-MBRL-AC, pretrained on six AC training tracks and PIA-MBRL-ODE, pretrained on the same six AC tracks plus three ODE datasets aligned with the held-out test tracks. Results in Table 2 and 3b are aggregated over lap times and failures rate. Detailed per-task results in the Appendix F.

IQL, a purely offline baseline, fails to adapt to unseen tasks, underscoring the limits of offline methods for generalization. SAC trained from scratch also struggles to adapt within the interaction budget. FOWM adapts more quickly, but remains suboptimal and less robust. In contrast, our methods outperform these baselines, with ODE augmentation yielding the strongest overall performance. The improvements are most evident on complex tracks and low-grip surfaces. Even on simpler tasks such as IND, where lap-time gains are minimal, ODE augmentation still improves safety by lowering incident rates.

*Table 1.* **Aggregated performance across tasks** Best lap time (s) for IQL, SAC scratch, FOWM, and physics-informed approaches. Mean of 3 seeds. **Lower is better** ($\downarrow$). Physics-informed augmentation (PIA-MBRL-ODE) consistently improves lap time over baseline.

| Task | IQL | SAC | FOWM | PIA-AC | PIA-ODE |
|------|------|------|------|------|------|
| IND | 159.33 | 60.00 | 60.19 | **59.97** | 60.00 |
| RBR | Fail | 255.64 | 87.83 | 85.85 | **84.34** |
| SIL | 209.95 | Fail | 118.88 | 114.16 | **112.17** |
| SIL$_{green}$ | Fail | Fail | 122.82 | 114.94 | **112.81** |
| SIL$_{dusty}$ | Fail | Fail | 125.85 | 119.79 | **117.61** |
| Average | 184.64 | 157.82 | 103.11 | 98.94 | **97.39** |

Figure 3a shows failure locations on RBR. Most crashes occur in the first corner after the starting position and in a complex corner later in the lap, indicating that the algorithm requires some interaction to adapt to new tracks and challenging maneuvers. Figure 3b demonstrates that PIA-MBRL-ODE accelerates adaptation, reducing the number of episodes needed to achieve the first successful episode. Offline pretraining provides a strong initialization, explaining why methods trained from scratch adapt more slowly. Overall, our framework consistently outperforms existing methods. Strong design choices such as the use of TD-MPC2 and planner regularization provide a robust foundation, but ODE augmentation is a key factor that expands dataset diversity, yielding more reliable performance in both nominal and challenging conditions.

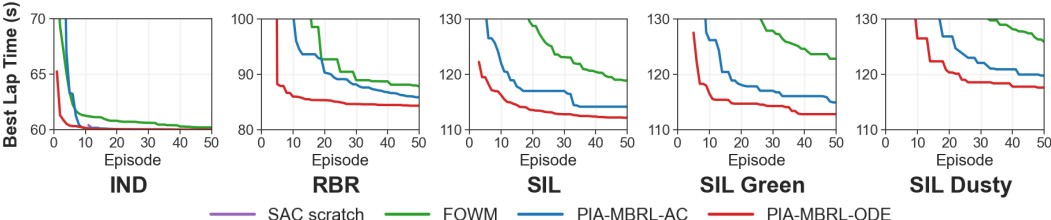

*Figure 4.* Best lap time (s) across different tracks and surface conditions. Physics-informed augmentation (PIA-MBRL-ODE) adapts faster and achieves lower lap times compared to baselines.

**Robustness to Domain Shift** Having established efficacy on nominal conditions, we test whether physics-informed augmentation improves adaptation under surface domain shifts. This is critical in racing, where grip varies with weather, and controllers that overfit to a single surface fail when conditions change.

Policies are pretrained on datasets collected under the optimal surface and fine-tuned on *dusty* and *green* surfaces, both of which reduce tire grip and make the car harder to control. Importantly, the ODE augmentation is generated using nominal vehicle parameters and does not include variations in tire friction or surface parameters. We did not experiment with varying ODE parameters across surfaces.

*Table 2.* **Aggregated failures across tasks.** Miles per failure across tracks for IQL, SAC scratch, FOWM, and our approaches. Mean of 3 seeds. **Higher is better** (↑). Physics-informed augmentation (PIA-MBRL-ODE) consistently improves safety over baselines.

| Task | SAC | FOWM | PIA-AC | PIA-ODE |
|---|---|---|---|---|
| IND | 80.73 | 714.68 | 1194.29 | **1233.25** |
| RBR | 0.39 | **39.14** | 26.40 | 34.48 |
| SIL | 0.28 | 32.06 | 34.62 | **68.80** |
| $SIL_{green}$ | 0.02 | 19.54 | 42.68 | **46.57** |
| $SIL_{dusty}$ | 0.02 | 16.50 | 16.48 | **25.57** |
| Average | 16.29 | 164.38 | 262.09 | **281.73** |

Results for the SIL, $SIL_{green}$, and $SIL_{dusty}$ conditions are shown in Tables 2 and 1, as well as Figure 4. We observe that all methods degrade relative to the optimal-surface baseline, with SAC failing entirely under these conditions. In contrast, ODE-augmented TD-MPC2 maintains a clear advantage, achieving competent adaptation with fewer crashes. These results demonstrate that physics-informed augmentation improves robustness even without explicitly modeling surface variation. The fixed ODE parameters provide consistent geometric and dynamical priors that transfer across conditions, but they also highlight potential for improvement: allowing ODE parameters to vary with grip differences could further enhance adaptation. This capability would be particularly valuable in applications such as street driving, where weather and surface conditions change frequently.

**Scaling** A natural question is how the benefits of our approach interact with dataset diversity. In RL, larger and more varied offline datasets are known to support better generalization, but this relationship has not been studied in the context of high-speed racing. We therefore ask: *does increasing the number of training tracks improve adaptation to unseen conditions?*

To address this, we pretrain on AC datasets with 1T, 3T, 6T, 9T, and 12T splits, then fine-tune on the test tracks, keeping the adaptation budget fixed. As shown in Figure 5, policies benefit substantially from increased dataset diversity. Larger and more varied pretraining sets yield faster convergence,

lower lap times, and fewer incidents. In particular, policies pretrained on 12T adapt most rapidly and achieve the strongest final lap times on the test tracks.

These findings mirror trends in RL more broadly: greater diversity in offline experience produces more robust latent dynamics and planners, reducing sensitivity to distribution shift. In our setting, this diversity comes from the inclusion of more tracks with distinct layouts. Here we isolate the effect of dataset diversity using only AC data, showing that it is key for adaptation. In the following experiment, we examine how ODE provides a cost-effective way to supply this diversity when AC data is limited.

**Synthetic-only Pretraining** We now investigate whether purely synthetic rollouts can replace, or at least complement, high-fidelity data when the latter is scarce. This is particularly relevant in autonomous racing, where real-world data collection is costly and dangerous. We therefore ask: can policies pretrained exclusively on synthetic rollouts achieve competitive adaptation, and can a small amount of simulator data of the target domain serve to anchor this pretraining? We compare five variants: **SCRATCH** (PIA-MBRL with no pretraining), **ODE ONLY** (pretrained solely on ODE rollouts, 9ODE+3ODE), **AC ONLY** (pretrained solely on simulator rollouts in the target domain, PIA-AC), **LOW ODE DATA** (PIA-ODE, pretrained on ODE rollouts from three target tracks), and **LOW AC DATA** (a mixed setting with 1M AC steps on a single track plus ODE rollouts from three tasks, 1M1T+3ODE). Figure 6 shows the effect of varying the proportion of synthetic data and target simulator data in the offline buffer. First, training solely on ODE rollouts already produces a competitive policy, significantly outperforming scratch training under the small adaptation budget. Second, seeding even a small number of real samples into a large synthetic dataset dramatically boosts performance, highlighting the value of mixed pretraining strategies. Finally, while synthetic-only pretraining is viable when no high-fidelity data is available, the best overall results consistently come from combining synthetic and real data, suggesting that physics-informed augmentation is most powerful as a bridge rather than a full replacement.

**Ablations** We study the role of ODE signals, planner regularization, Task and Context IDs. Results show that geometry signals aid safe exploration, regularization stabilizes adaptation, and task IDs improve generalization. Extended analyses are provided in the appendix.

## 5 RELATED WORK

**Model-based augmentation.** Since Dyna-Q (Sutton, 1990), synthetic rollouts from learned dynamics have been widely explored (e.g., MBPO (Janner et al., 2019), MOPO (Yu et al., 2020), Dreamer (Hafner et al., 2019), and TD-MPC2 (Hansen et al., 2023)), and have shown that model-based rollouts can significantly improve sample efficiency. However, they are prone to compounding error, so rollouts are typically kept short to remain reliable. Physics-informed models such as PhIHP (Asri et al., 2024) mitigate this by combining analytic priors with

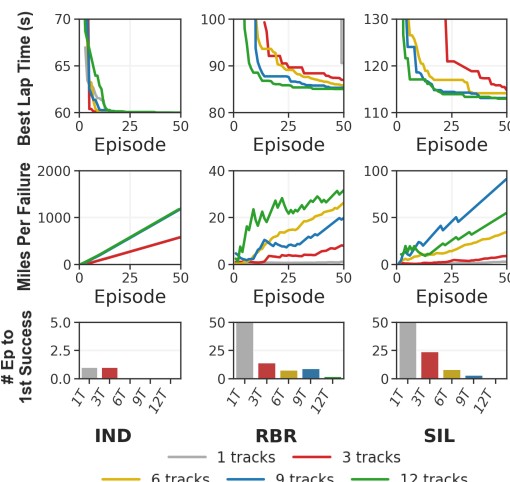

*Figure 5.* **Scaling.** Pretraining on more tracks yields faster adaptation, lower lap times, and fewer incidents when fine-tuning on test tracks.

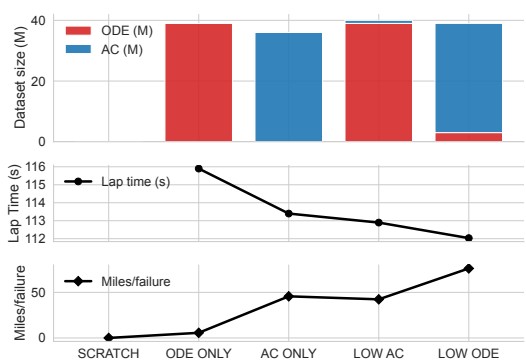

*Figure 6.* **Synthetic-only Pretraining.** Policies pretrained on synthetic rollouts alone already achieve competitive adaptation, while mixing in a small amount of real data yields the best results.

residual corrections, but are still constrained to short rollouts. Djeumou et al. (2024) trains a neural SDE from real data to pretrain policies for zero-shot sim-to-real transfer, and AnyCar (Xiao et al., 2024) similarly explores the use of transformers for world model generation and zero-shot deployment. While impressive, these methods pretrain and deploy policies with learned models, and are limited by model fidelity and the absence of online adaptation. In contrast, we use a fixed analytical ODE to generate stable, long-horizon rollouts, merge them with high-fidelity simulator data, and train using state of the art MBRL methods. This shifts the paradigm from online learning with learned world models to offline augmentation with analytic models, enabling broader coverage and efficient adaptation. Unlike imitation-based methods such as GPS (Levine & Koltun, 2013), we use analytic models as data generators rather than teachers, with MBRL bridging synthetic and real data.

**Offline RL and adaptation.** Offline RL methods such as IQL (Kostrikov et al., 2021) enforce conservatism to mitigate value overestimation, but this limits generalization outside the dataset's support. H2O (Niu et al., 2022) extends IQL by mixing offline data with simulator rollouts and down-weighting out-of-distribution samples, but still emphasizes conservatism. Offline-to-online approaches like FOWM (Feng et al., 2023) show that offline pretraining can speed up adaptation with limited interaction. We extend this line by using SOTA MBRL together with ODE augmentation and key design choices, yielding safer and more efficient generalization to unseen tasks.

**Autonomous racing.** MPC-based controllers deliver strong performance but require heavy modeling (Betz et al., 2023; Raji et al., 2022; 2024; Wischnewski et al., 2023), while RL systems such as GT Sophy (Wurman et al., 2022) and AssettoCorsaGym (Remonda et al., 2024) achieve human-level results with massive interaction budgets. Subosits et al. (2024) train MFRL with domain randomization over grip setups to achieve robustness. In contrast, we evaluate generalization without explicit randomization, showing that robustness emerges from ODE augmentation + MBRL adaptation on unseen tracks and surfaces.

## 6 DISCUSSION AND CONCLUSION

Our study shows that physics-informed augmentation with lightweight models improves sample efficiency, safety, and generalization in model-based RL. Reliable rollouts from analytical models provide a scalable alternative to learned dynamics, reframing augmentation as simulation rather than model learning. Larger and more diverse pretraining buffers further enhance adaptation, while synthetic-only pretraining demonstrates potential in data-scarce or unsafe domains. Mixing synthetic and simulator data yields the strongest performance overall.

While augmentation provides structured priors, it does not capture all real-world complexities. Our robustness experiments show that adaptation to unseen surfaces improves over baselines, but performance still degrades relative to the nominal case. Extending the framework to include parameterized families of ODE models or hybrid neural–analytical models may enhance robustness. Another limitation is the evaluation domain: although autonomous racing provides an extreme and safety-critical setting, further validation in robotics, UAV control, and locomotion is needed to establish generality.

Finally, we note that beyond improving generalization, this approach opens possibilities for more complex scenarios. In autonomous racing, augmentation could help agents generalize to rare but critical maneuvers such as overtaking, coordinating with teammates, or executing long-horizon race strategies that are difficult to capture in demonstration data. Similarly, in domains like robotic manipulation, UAVs, or locomotion, lightweight models could be used to pretrain agents for behaviors that are rarely observed or unsafe to attempt directly, providing a structured foundation for safer and more efficient exploration.

In conclusion, we introduced PIA-MBRL, a framework that leverages physics-informed data augmentation with lightweight models to improve adaptation and generalization in high-performance control. By combining stable analytical rollouts with scalable MBRL, the method achieves rapid adaptation across diverse tracks and surfaces in autonomous racing, a setting where both safety and efficiency are of maximum importance. Beyond racing, the framework is applicable to any domain where lightweight analytical models exist, offering a practical path toward safer, more efficient reinforcement learning in real-world systems.

## STATEMENTS

**Reproducibility statement** We are committed to ensuring reproducibility of our experiments and will open-source code for training and evaluation of policies upon acceptance. We provide detailed information about our experimental setup, including the specific tracks that we use (Appendix B.1), our environment interface (Appendix B.3), vehicle specification (Appendix B.4), the full ODE vehicle model used (Appendix C), implementation details and hyperparameters for our method and baselines (Appendix G), as well as full, de-aggregated results throughout the appendices.

**Details of LLM use** We used large language models to aid and polish writing. For writing assistance, the authors drafted the text and used model suggestions to improve clarity, grammar, and flow. All text was reviewed and edited by the authors; the models did not originate technical ideas, methods, results, or claims. No experimental design or data analysis was produced by LLMs.

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

APPENDIX

## A   SYSTEM IMPROVEMENTS

We make following engineering system improvements based on the ACGYM framework to enable large-scale pretraining and real-time finetuning:

- **Pluggable Dynamics:** Refactored the environment to support swapping different dynamics models (high-fidelity AC vs. lightweight ODE);

- **Real-time TD-MPC2:** Optimized JAX inference to satisfy real-time constraints for MBRL;

- **Automation:** Scripted track/surface and car setups swappings for automated experiments on Linux;

- **Others (Optional):** Support for policy gear shifting and image capture.

## B   ASSETTO CORSA ENVIRONMENT

Assetto Corsa environment provides a diverse collection of racing circuits and surface conditions, serving as the high-fidelity backbone for our experiments.

### B.1   TRACKS

We include 15 tracks spanning a wide range of layouts, geographies, and difficulty levels (see Figure 7):

- Barcelona, Spain (BRN)

- Monza, Italy (MNZ)

- Shanghai International Circuit, China (CHI)

- Imola, Italy (IML)

- Monaco, Monaco (MON)

- Laguna Seca, USA (LAS)

- Suzuka, Japan (SUK)

- Tsukuba, Japan (TSU)

- Road America, USA (RAM)

- Zandvoort, Netherlands (NED)

- Spa-Francorchamps, Belgium (SPA)

- Bathurst (Mount Panorama), Australia (AUT)

- Silverstone, United Kingdom (SIL)

- Red Bull Ring, Austria (RBR)

- Indianapolis Motor Speedway, USA (IND)

*Figure 7.* Layout of the 15 tracks used in our experiments. The first 12 tracks are included in the training dataset, while the remaining three (RBR, SIL, and IND) are reserved for testing.

## B.2 SURFACES

We consider three surface conditions provided by Assetto Corsa, each corresponding to a different grip setting:

- **Optimal**: Maximum grip is applied across the track ($\approx 100\%$ grip multiplier). This represents a perfectly rubbered-in racing surface with stable conditions, offering the highest traction and fastest lap times.

- **Green**: A freshly cleaned track with minimal rubber deposited ($\approx 94\%$ grip multiplier). Grip gradually increases as more laps are driven, simulating the natural evolution of a racing weekend.

- **Dusty**: A low-grip configuration ($\approx 86\%$ grip multiplier) that introduces additional sliding and instability. This setting simulates tracks with dust, dirt, or lack of rubber, making vehicle control significantly harder.

### B.3 ENVIRONMENT INTERFACE

The Assetto Corsa environment is interacted with a `gym`-compliant API, enabling straightforward integration with standard RL algorithms. The interface defines the observation space, action space, reward function, and termination criteria as follows.

**Observations.** The environment is partially observable. At each step, agents receive an asynchronous state vectors $\mathcal{S} \in \mathbb{R}^{125}$ generated from the most recent telemetry information from Assetto Corsa. To mitigate partial observability, the state representation stacks telemetry from the three most recent timesteps and appends the absolute control inputs from the previous two actions. Following Remonda et al. (2021), the telemetry includes linear and angular velocities and accelerations, range finder distances to track edges, look-ahead curvature, steering wheel force feedback (a proxy for tire grip used by human drivers), the angle between wheel orientation and motion direction, and the 2D distance to a reference racing line provided by AC. The reference line is a fixed, car-independent trajectory, while the true optimal line depends on the vehicle, so agents can benefit from deviating when advantageous.

**Actions.** The continuous action space $\mathcal{A} \in \mathbb{R}^3$ includes throttle, brake, and steering inputs, each normalized to $[-1, 1]$ for integration with RL algorithms. These commands are mapped to the simulator's maximum control ranges. To reduce oscillations, controls are applied as deltas rather than absolutes. An automated gearbox is used throughout.

**Reward.** The reward is defined as the product of the vehicle's forward velocity $v$ and a penalty term for lateral deviation $d$ from the reference line:

$$ r = v \cdot (1 - \alpha d), $$

where $\alpha$ is a tunable coefficient controlling the strength of line-following supervision. Higher values of $\alpha$ encourage strict adherence to the reference path, while lower values allow more flexibility to explore alternative racing lines.

**Episode termination.** An episode terminates early if the car leaves the track with three or more wheels beyond the boundary, or if its speed falls below 5 km/h for more than two seconds.

### B.4 VEHICLE SPECIFICATIONS

We consider a single car:

**Formula 3:** Formula 3 car in line with FIA Formula 3 regulations. This is of mid difficulty with a top speed of approximately 250 km/h. This model is contributed by the community (https://www.racedepartment.com/downloads/rsr-formula-3.8040/).

The car weighs about 570 Kg and is powered by a 246 bhp engine producing 320 Nm of torque. It reaches a top speed of around 230 km/h, with a weight-to-power ratio of approximately 2.03 kg/bhp. Other parameters are listed in Table 3:

*Table 3.* Vehicle and drivetrain parameters from Assetto Corsa configs.

| Group | Parameter | Symbol | Value | Unit |
|---|---|---|---|---|
| Engine | Idle / limiter | $\mathrm{rpm}_{\min}/\mathrm{rpm}_{\max}$ | 1700 / 7500 | rpm |
| Drivetrain | Gear ratios (0–6) | $g_i$ | [2.75, 2.07, 1.667, 1.37, 1.17, 1.0] | – |
| Drivetrain | Final drive | FD | 2.833 | – |
| Drivetrain | Rear wheel radius | $R_w$ | 0.2785 | m |
| Chassis | Mass | $m$ | 570 | kg |
| Chassis | Yaw inertia | $I_z$ | 1500 | $\mathrm{kg\,m}^2$ |
| Gravity | Gravity | $g$ | 9.81 | $\mathrm{m/s}^2$ |
| Geometry | CoG height | $h$ | 0.28 | m |
| Geometry | Front / rear CoG | $l_f/l_r$ | 1.652 / 1.148 | m |
| Brakes | Front / rear coeffs | $C_{Bf}/C_{Br}$ | 6300 / 4800 | – |
| Steering | Steering ratio | $\rho_\delta$ | 13.5 | – |

## C ODE MODELS

We develop a lightweight ODE vehicle model to generate synthetic rollouts for data augmentation. The model is designed to be lightweight, differentiable, and efficient enough for large-scale data generation, yet expressive enough (combined slip, aero, torque curve) to capture high-speed racing dynamics.

Similar to prior work acgym and Raji et al. (2023), we model a rear-wheel-drive race car as a 6-state, 4-input rigid-body bicycle model. The state is $x = [X, Y, \psi, v_x, v_y, r]^\top$, containing CoG position, yaw, body-frame velocities, and yaw rate. The controls are $u = [\delta_{\mathrm{cmd}}, D, B, G]^\top$, with steering command $\delta_{\mathrm{cmd}}$, throttle $D \in [0, 1]$, brake $B \in [0, 1]$, and gear $G$. The model assumes rear drive (with front braking), quadratic aerodynamic drag, and quadratic downforce with a fixed front split.

Vehicle parameters come from two sources. First, we directly import specifications from Assetto Corsa (AC), including mass, geometry, inertia, drivetrain ratios, and brake/steering coefficients (Table 3). Second, aerodynamic and tire parameters, which are not available in AC, are estimated from telemetry data (Table 4). Tire forces are represented with the Magic Formula (Pacejka, 1982). Hyperparameters were initialized from typical racing and then refined by system identification. Aerodynamic and chassis parameters were cross-checked against published specifications for formula cars in *Race Car Vehicle Dynamics* (Milliken & Milliken, 1994). We also used priors from related work Raji et al. (2023), before finalizing values through data-driven optimization.

The engine torque map (Table 5) is taken directly from AC and specifies the maximum torque at full throttle as a function of RPM. Continuous-time dynamics $\dot{x} = f(x, u)$ are integrated using a fixed-step RK4 scheme (50 Hz integration, 25 Hz control frequency).

### C.1 RIGID-BODY KINEMATICS AND DYNAMICS (BODY FRAME)

Kinematics:

$$\dot{X} = v_x \cos\psi - v_y \sin\psi, \qquad \dot{Y} = v_x \sin\psi + v_y \cos\psi, \qquad \dot{\psi} = r.$$

Equations of motion are:

$$\dot{v}_x = \frac{F_{\mathrm{drag}} + F_{xr} - F_{yf}\sin\delta + F_{xf}\cos\delta + mv_y r}{m},$$
$$\dot{v}_y = \frac{F_{yr} + F_{xf}\sin\delta + F_{yf}\cos\delta - mv_x r}{m},$$
$$\dot{r} = \frac{-F_{yr}l_r + (F_{xf}\sin\delta + F_{yf}\cos\delta)\, l_f}{I_z}.$$

Total accelerations:

$$a_y = v_x r + \dot{v}_y, \qquad a_x = \dot{v}_x - r v_y$$

### C.2 SLIP ANGLES

Steering at the tire is $\delta = \delta_{\mathrm{cmd}}/\mathrm{steer\_ratio}$. Using a standard bicycle model, slip angles are

$$\alpha_f = \arctan\!\left(\frac{v_y + r l_f}{v_x}\right) - \delta, \qquad \alpha_r = \arctan\!\left(\frac{v_y - r l_r}{v_x}\right).$$

### C.3 NORMAL LOADS: STATIC + AERODYNAMICS

With wheelbase $w_b = l_f + l_r$,

$$F_{zf0} = mg\frac{l_r}{w_b}, \qquad F_{zr0} = mg\frac{l_f}{w_b}.$$

Aero downforce with coefficient $C_L$, area $S$, density $\rho$, and front split $\eta$:

$$F_{zf}^{\text{aero}} = \tfrac{1}{2}\rho v_x^2 C_L S\, \eta, \qquad F_{zr}^{\text{aero}} = \tfrac{1}{2}\rho v_x^2 C_L S\, (1 - \eta),$$

and total (nonnegative) loads

$$F_{zf} = \max(0, F_{zf0} + F_{zf}^{\text{aero}}), \quad F_{zr} = \max(0, F_{zr0} + F_{zr}^{\text{aero}}).$$

## C.4 LATERAL TIRE FORCES (PACEJKA-STYLE)

Per axle, a simplified Pacejka form with identified $(B, C, D, E, S_h, S_v)$, scaled by normal load:

For each axle $i \in \{f, r\}$ with slip $\alpha_i$ and load $F_{zi}$:

$$\phi_i = B_i(\alpha_i + S_{h,i}), \quad F_{yi}^{\text{raw}} = F_{zi}\big(S_{v,i} + D_i \sin\big(C_i \arctan \phi_i\big) - E_i\big(\phi_i - \arctan \phi_i\big)\big).$$

## C.5 DRIVETRAIN AND ENGINE TORQUE

Overall ratio: $\tau = \texttt{gear\_ratio}[g] \cdot \texttt{final\_ratio}$, with $g = \text{round}(g_f)$. Engine speed (rpm) from kinematics:

$$\text{rpm} = \text{clip}\Big(\tfrac{v_x \tau}{R_w} \cdot \tfrac{60}{2\pi}, \text{RPM}_{\min}, \text{RPM}_{\max}\Big)$$

Where the factor $\tfrac{60}{2\pi}$ converts rad/s $\rightarrow$ rpm.

Engine torque and drive force:

$$T_{\text{eng}} = T(\text{rpm}) \cdot D, \qquad F_{\text{eng}} = \tfrac{T_{\text{eng}} \tau}{R_w}.$$

## C.6 LONGITUDINAL FORCES AND DRAG

$$F_{\text{drag}} = -\tfrac{1}{2}\rho v_x^2 C_D S\, \mathbf{1}[v_x > 0],$$
$$F_{xf} = -C_{Bf} B\, \mathbf{1}[v_x > 0],$$
$$F_{xr} = F_{\text{eng}} - C_{Br} B\, \mathbf{1}[v_x > 0].$$

Pure-longitudinal capacity limits:

$$F_{xf} \in [-D_{fx} F_{zf}, 0], \qquad F_{xr} \in [-D_{rx} F_{zr}, D_{rx} F_{zr}].$$

## C.7 COMBINED-SLIP SATURATION (FRICTION ELLIPSE)

Define normalized demands

$$k_{x,f} = \frac{F_{xf}}{D_{fx} F_{zf} + \varepsilon}, \quad k_{y,f} = \frac{F_{yf}}{D_{f\ell} F_{zf} + \varepsilon},$$

(and analogously for the rear). If $\sqrt{k_x^2 + k_y^2} > 1$, scale both components uniformly by

$$G = \frac{1}{\sqrt{k_x^2 + k_y^2}} \quad \Rightarrow \quad (F_x, F_y) \leftarrow G\,(F_x, F_y).$$

This yields smooth axle-wise combined-slip saturation while preserving force direction.

The engine torque map is taken directly from the AC configuration. It specifies the maximum engine torque at full throttle as a function of rpm, with a peak of about 320 N·m at 5000 rpm.

*Table 4.* Parameters estimated via system identification (initialized from typical values).

| Group | Parameter | Symbol | Value | Unit |
|---|---|---|---|---|
| Aero | Drag coefficient | $C_D$ | 0.95 | – |
| Aero | Downforce coefficient | $C_L$ | 3.49 | – |
| Aero | Air density | $\rho$ | 1.2 | kg/m$^3$ |
| Aero | Ref. area | $S$ | 1.0 | m$^2$ |
| Aero | Front downforce split | $\eta$ | 0.413 | – |
| Tire (front) | Pacejka $B, C, D, E, S_h, S_v$ | – | 14.368, 1.38, 1.242, 0, 0, 0 | – |
| Tire (rear) | Pacejka $B, C, D, E, S_h, S_v$ | – | 16.15, 0.9793, 1.6896, $-0.5$, 0, 0 | – |
| Longitudinal | Capacity scale (front) | $D_f^x$ | 5.6 | – |
| Longitudinal | Capacity scale (rear) | $D_r^x$ | 1.2 | – |

*Table 5.* Engine torque map at full throttle provided by Assetto Corsa.

| RPM | Torque [N m] |
|---|---|
| 0 | 50 |
| 500 | 70 |
| 1000 | 78 |
| 1500 | 100 |
| 2000 | 111 |
| 2500 | 144 |
| 3000 | 189 |
| 3500 | 239 |
| 4000 | 302 |
| 4500 | 311 |
| 5000 | 320 |
| 5500 | 316 |
| 6000 | 292 |
| 6500 | 262 |
| 7000 | 238 |
| 7500 | 218 |

## D DATASET

We provide a comprehensive dataset for autonomous racing, including over 740 hours of driving data collected by running SAC in Assetto Corsa at 25 Hz on a workstation with an NVIDIA RTX 3070 GPU, together with synthetic rollouts generated using SAC in physics-based ODE vehicle models at 98 Hz on an NVIDIA RTX 5080 GPU. Detailed statistics are provided in Table 6.

*Table 6.* Dataset statistics for Assetto Corsa (AC) and ODE environments. AC data are collected at real-time speed (25Hz), while ODE rollouts run nearly four times faster (average 98Hz), enabling large-scale synthetic data generation.

| Source | Track | Data Size (# steps) | GPU Hours (h) | Sampling Efficiency (Hz) |
|---|---|---|---|---|
| | BRN | 4 M | 44.9 | 25 |
| | MNZ | 2.5 M | 28.1 | 25 |
| | CHN | 6 M | 67.2 | 25 |
| | MON | 6 M | 67.2 | 25 |
| | IML | 6 M | 67.1 | 25 |
| Assetto | LAS | 6 M | 67.1 | 25 |
| Corsa | SUK | 6 M | 67.1 | 25 |
| | TSU | 6 M | 67.1 | 25 |
| | RAM | 6 M | 67.1 | 25 |
| | ZAN | 6 M | 67.1 | 25 |
| | SPA | 6 M | 67.8 | 25 |
| | NED | 6 M | 67.2 | 25 |
| | SIL | 6 M | 17.2 | 96.9 |
| ODE | RBR | 6 M | 15.9 | 104.8 |
| | IND | 6 M | 17.7 | 94.2 |

## E    TRAINING AND EVALUATION

**Training Setup**    All methods follow the same offline $\to$ online protocal.

Offline pretraining is conducted on a $8 \times$ NVIDIA Geforce RTX 3090 GPU server, while online finetuning is performed on a NVIDIA GeForce RTX 4090 GPU workstation to meet the real-time requirements of ACGym. After each online episode, model updates with batches that mix offline and online transitions in a 50/50 ratio.

Online finetuning in Assetto Corsa proceeds over multiple episodes, where each consists of about 7 track laps (depending on track and car), within $0.1h$ wall clock time (at 25 Hz). Episodes terminate until 15,000 steps are reached or when the car stalls or gets out of the track.

**Evaluation Metrics**    To capture the driving performance, learning efficiency and safety, we evaluate performance of the model using the following metrics:

- **Best Lap Time** ↓: minimum lap time achieved during model performances.

- **Best Reward** ↑: maximum episodic return.

- **# Episodes to 1st Success** ↓: number of episodes required to complete the first valid lap.

- **# Crashes** ↓: count of hard incidents (collisions/off-track).

- **# Low Speed** ↓: count of stall events (speed $< 5$ km/h for $> 2$s).

- **# Failures** ↓: total count of failing episodes (# Crashes + # Low Speed).

- **Miles per Failure** ↑: total driving distance divided by total count of failures.

We regard an algorithm as better when it achieves faster lap times, demonstrates fast learning efficiency (i.e. fewer # Episodes to 1st Success) and ensures safe driving (i.e. fewer number of failures).

## F    EXPERIMENT RESULTS

In this section, we provide detailed results with regards to each experiment and ablation.

## F.1  EXPERIMENT 1 & 2

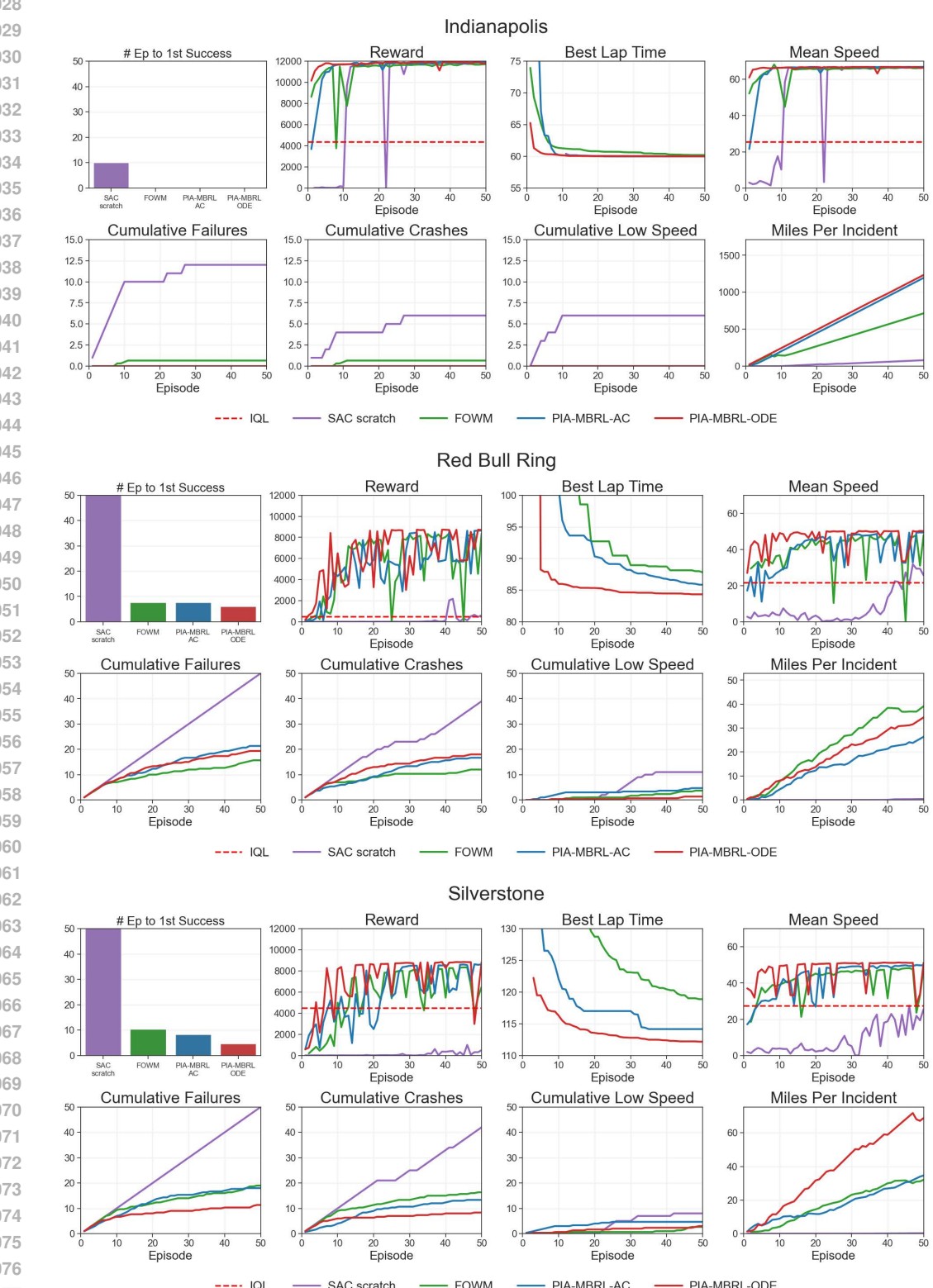

*Figure 8.* Online performances results: Indianapolis (top), Red Bull Ring (middle), and Silverstone (bottom).

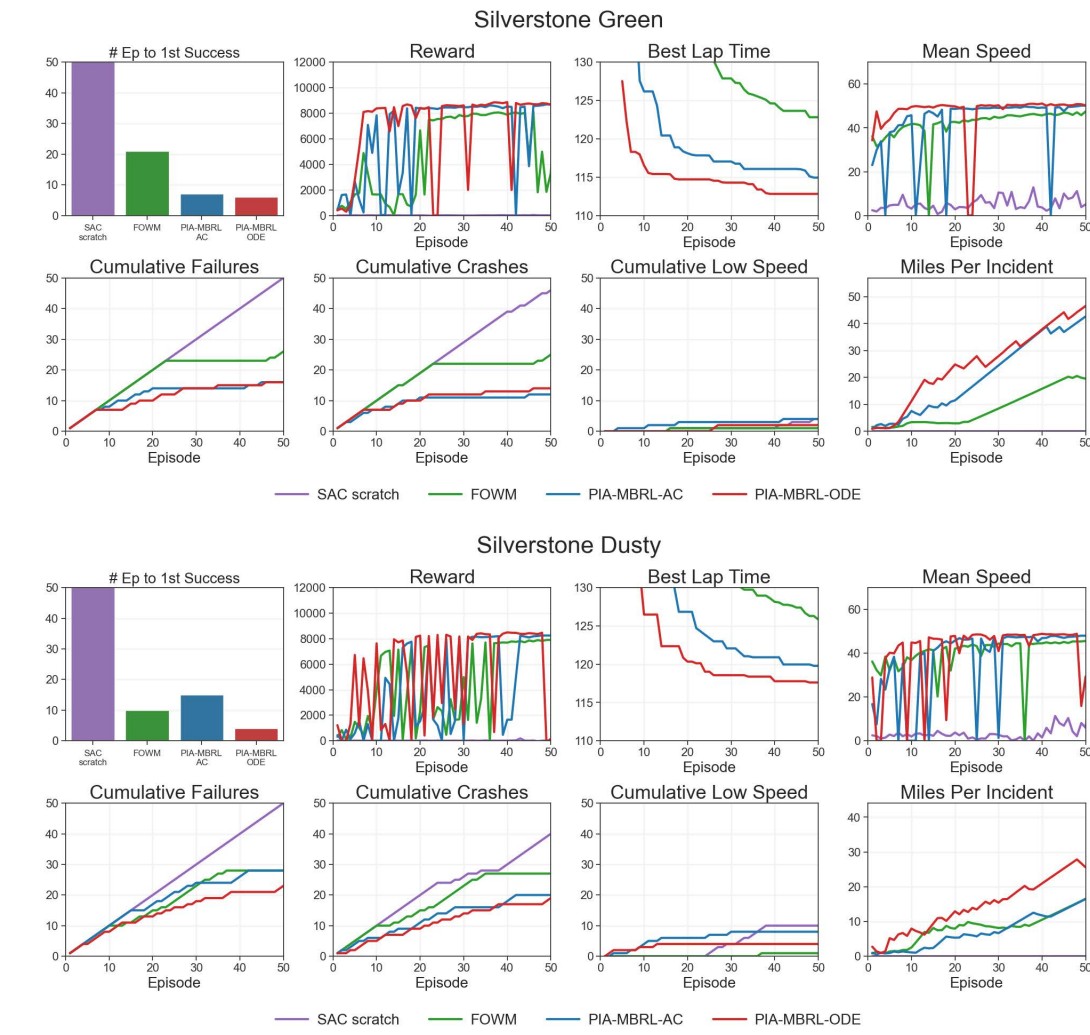

*Figure 9.* Online finetuning results: Silverstone Green Surface (top) and Silverstone Dusty Surface (bottom).

*Table 7.* Online performance results across tracks and surface conditions. Physics-informed augmentation (PIA-MBRL-ODE) achieves better lap time with fewer # Ep to 1st Success and failures, demonstrating better lap time, safety and learning efficiency than other baselines across different tracks and surface conditions.

| Model | Best Lap (s) ↓ | Best Reward ↑ | # Ep to 1st Success ↓ | # Failures ↓ | # Crashes ↓ | # Low Speed ↓ | Miles per Failure ↑ |
|---|---|---|---|---|---|---|---|
| **Indianapolis (Optimal Surface)** | | | | | | | |
| IQL | 159.33 | 4350.30 | – | – | – | – | – |
| SAC scratch | 60.00 | 11923.34 | 10.00 | 12 | 6 | 6 | 80.73 |
| FOWM | 60.19 | 11784.19 | 0.00 | 0 | 0 | 0 | 714.68 |
| PIA-MBRL-AC | **59.97** | **11944.55** | 0.00 | 0 | 0 | 0 | 1194.29 |
| PIA-MBRL-ODE | 60.00 | 11938.17 | 0.00 | 0 | 0 | 0 | **1233.25** |
| **Red Bull Ring (Optimal Surface)** | | | | | | | |
| IQL | Fail | 511.67 | – | – | – | – | – |
| SAC scratch | 255.64 | 2176.23 | Fail | 50 | 39 | 11 | 0.39 |
| FOWM | 87.83 | 8378.39 | 7.67 | **15** | **12** | 3 | **39.14** |
| PIA-MBRL-AC | 85.85 | 8641.52 | 7.67 | 21 | 17 | 4 | 26.40 |
| PIA-MBRL-ODE | **84.34** | **8734.56** | **6.00** | 19 | 18 | **1** | 34.48 |
| **Silverstone (Optimal Surface)** | | | | | | | |
| IQL | 209.95 | 4506.50 | – | – | – | – | – |
| SAC scratch | Fail | 989.71 | Fail | 50 | 42 | 8 | 0.28 |
| FOWM | 118.88 | 8335.25 | 10.33 | 19 | 17 | **2** | 32.06 |
| PIA-MBRL-AC | 114.16 | 8637.48 | 8.33 | 18 | 14 | 4 | 34.62 |
| PIA-MBRL-ODE | **112.17** | **8852.00** | **4.67** | **11** | **8** | 3 | **68.80** |
| **Silverstone (Green Surface)** | | | | | | | |
| IQL | Fail | 373.58 | – | – | – | – | – |
| SAC scratch | Fail | 39.02 | Fail | 50 | 46 | 4 | 0.02 |
| FOWM | 122.82 | 8147.35 | 21 | 26 | 25 | **1** | 19.54 |
| PIA-MBRL-AC | 114.94 | 8691.82 | 7 | 16 | **12** | 4 | 42.68 |
| PIA-MBRL-ODE | **112.81** | **8869.60** | **6** | 16 | 14 | 2 | **46.57** |
| **Silverstone (Dusty Surface)** | | | | | | | |
| IQL | Fail | 374.04 | – | – | – | – | – |
| SAC scratch | Fail | 183.22 | Fail | 50 | 40 | 10 | 0.02 |
| FOWM | 125.85 | 7906.53 | 10 | 28 | 27 | **1** | 16.50 |
| PIA-MBRL-AC | 119.79 | 8252.75 | 15 | 28 | 20 | 8 | 16.48 |
| PIA-MBRL-ODE | **117.61** | **8492.57** | **4** | **23** | **19** | 4 | **25.57** |

## F.2 EXPERIMENT 3

*Figure 10.* Data scalability results: Indianapolis (top), Red Bull Ring (middle), and Silverstone (bottom).

*Table 8.* Data scalability across tracks for three circuits. Training on a wider variety of tracks achieves better lap time speed and lowers failure rates with faster adaptation speed on test tracks.

| Indianapolis | | | | | | | |
|---|---|---|---|---|---|---|---|
| Model | Best Lap (s) ↓ | Best Reward ↑ | # Ep to 1st Success ↓ | # Failures ↓ | # Crashes ↓ | # Low Speed ↓ | Miles per Failure ↑ |
| 1 tracks | **59.91** | 11896.40 | 1 | 1 | 1 | 0 | 587.18 |
| 3 tracks | 60.01 | **11956.10** | 1 | 1 | 1 | 0 | 580.57 |
| 6 tracks | 59.97 | 11944.50 | 0 | 0 | 0 | 0 | **1194.29** |
| 9 tracks | 59.96 | 11942.70 | 0 | 0 | 0 | 0 | 1183.60 |
| 12 tracks | 60.01 | 11937.90 | 0 | 0 | 0 | 0 | 1175.30 |
| **Red Bull Ring** | | | | | | | |
| Model | Best Lap (s) ↓ | Best Reward ↑ | # Ep to 1st Success ↓ | # Failures ↓ | # Crashes ↓ | # Low Speed ↓ | Miles per Failure ↑ |
| 1 tracks | 90.59 | 5928.99 | Fail | 50 | 38 | 12 | 1.09 |
| 3 tracks | 87.01 | 8417.39 | 14 | 38 | 30 | 8 | 7.97 |
| 6 tracks | 85.85 | 8641.52 | 8 | 21 | 17 | **4** | 26.40 |
| 9 tracks | 85.32 | 8609.08 | 9 | 26 | 21 | 5 | 19.82 |
| 12 tracks | **85.06** | **8739.13** | **2** | **20** | **12** | 8 | **31.66** |
| **Silverstone** | | | | | | | |
| Model | Best Lap (s) ↓ | Best Reward ↑ | # Ep to 1st Success ↓ | # Failures ↓ | # Crashes ↓ | # Low Speed ↓ | Miles per Failure ↑ |
| 1 tracks | 142.72 | 4358.01 | Fail | 50 | 27 | 23 | 0.83 |
| 3 tracks | 114.81 | 8578.76 | 24 | 39 | 32 | 7 | 9.01 |
| 6 tracks | 114.16 | 8637.48 | 8 | 18 | 14 | 4 | 34.62 |
| 9 tracks | 113.13 | 8829.61 | 3 | **9** | 8 | **1** | **91.38** |
| 12 tracks | **112.99** | **8913.49** | **0** | 13 | **7** | 6 | 55.06 |

## F.3 EXPERIMENT 4

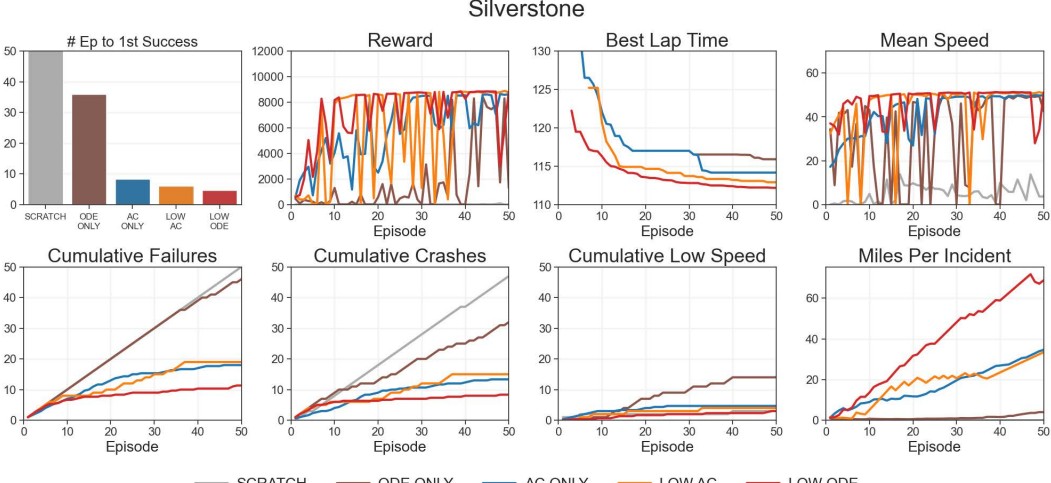

*Figure 11.* Synthetic-Only pretraining.

| Model | Best Lap (s) | Best Reward | # Ep to 1st Success | # Failures | # Crashes | # Low Speed | Miles per Failure |
|---|---|---|---|---|---|---|---|
| SCRATCH | Fail | 86.08 | – | 50 | 47 | 3 | 0.03 |
| ODE ONLY | 115.90 | 8441.36 | 36 | 46 | 32 | 14 | 3.98 |
| AC ONLY | 114.16 | 8637.48 | 8 | 18 | 13 | 4 | 34.62 |
| LOW AC | 112.90 | 8884.29 | 6 | 19 | 15 | 4 | 33.46 |
| LOW ODE | 112.17 | 8852.00 | 5 | 11 | 8 | 3 | **68.80** |

*Table 9.* Synthetic-Only pretraining. Pretraining on synthetic rollouts alone enables competitive adaptation, and adding a small amount of real data achieves the best results.

### F.4 ABLATIONS

To understand how individual components contribute to the success of our method, we conduct a series of ablations. We highlight key findings, with extended details in the appendix.

**ODE Signal ablations** We seek to determine whether ODE pretraining helps via geometry (spatial layout: track borders, curvature look-ahead), dynamics (proprioceptive signals), or both. We compare pretraining settings to isolate their contributions. We pretrain with: No-ODE (PIA-MBRL-AC), ODE (G+D) (PIA-MBRL-ODE), ODE G-only (mask dynamics on ODE samples), ODE D-only (mask geometry on ODE samples) and fine-tune on SIL. Geometry-only reduces incidents relative to No-ODE, confirming that spatial priors improve safe exploration, but has slower lap time. Dynamics-only underperforms on both metrics, while G+D is best on both.

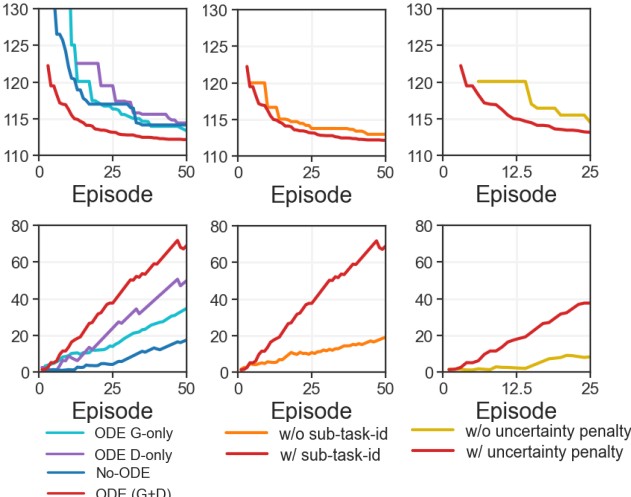

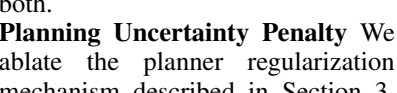

*Figure 12.* **Ablation studies.** Left: effect of geometry vs. dynamics signals in ODE pretraining; Middle: effect of conditioning on sub-task IDs; Right: effect of Uncertainty penalty.

**Planning Uncertainty Penalty** We ablate the planner regularization mechanism described in Section 3. Removing it leads to unstable adaptation and more failures. Regularization stabilizes early adaptation and ensures safer online data collection.

**Task and Context IDs** We ablate conditioning the policy on data source ID. Removing these inputs leads to worse generalization and noisier adaptation, confirming that explicit context helps disambiguate offline datasets.

### F.5 ABLATIONS - EXTRA RESULTS

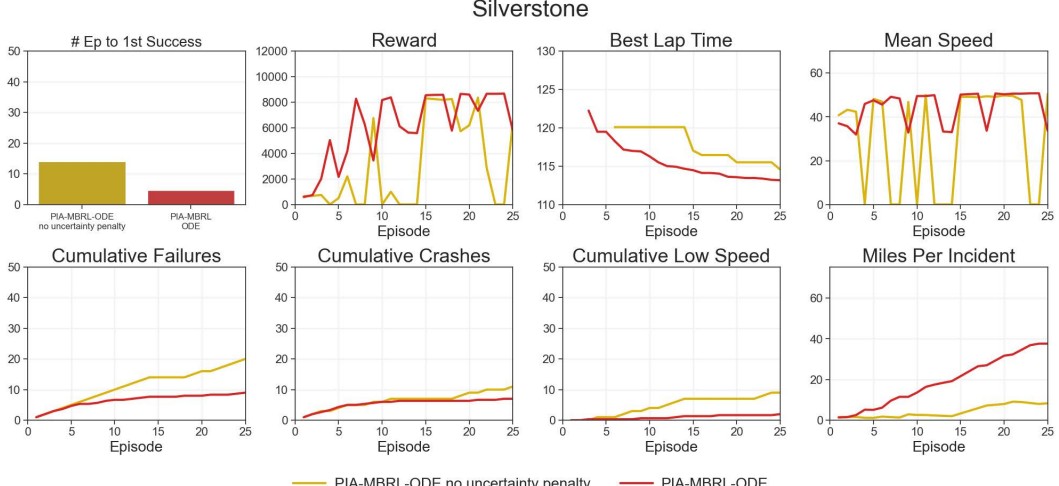

*Figure 13.* Ablation results: ODE Signal (top), Sub-Task ID (middle), and Planning Uncertainty Penalty (bottom).

| ODE Signal Ablations | | | | | | | |
|---|---|---|---|---|---|---|---|
| Model | Best Lap (s) ↓ | Best Reward ↑ | # Ep to 1st Success ↓ | # Failures ↓ | # Crashes ↓ | # Low Speed ↓ | Miles per Failure ↑ |
| No ODE (AC) | 114.16 | 8637.48 | 8 | 18 | 14 | 4 | 34.62 |
| ODE G only | 113.39 | 8735.38 | 9 | 14 | **8** | 6 | 49.72 |
| ODE D only | 114.43 | 8618.67 | 12 | 27 | 19 | 8 | 17.46 |
| ODE G+D | **112.17** | **8852.00** | **5** | **11** | **8** | **3** | **68.80** |
| Ablation on Sub-Task ID | | | | | | | |
| Model | Best Lap (s) ↓ | Best Reward ↑ | # Ep to 1st Success ↓ | # Failures ↓ | # Crashes ↓ | # Low Speed ↓ | Miles per Failure ↑ |
| W/o sub-task id | 112.98 | 8743.11 | 13 | 29 | 24 | 5 | 19.12 |
| W/ sub-task id | **112.17** | **8852.00** | **5** | **11** | **8** | **3** | **68.80** |
| Ablation on Planning Uncertainty Penalty | | | | | | | |
| Model | Best Lap (s) ↓ | Best Reward ↑ | # Ep to 1st Success ↓ | # Failures ↓ | # Crashes ↓ | # Low Speed ↓ | Miles per Failure ↑ |
| W/o uncertainty | 114.54 | 8364.59 | 14 | 20 | 11 | 9 | 8.32 |
| W/ uncertainty | **113.16** | **8677.11** | **5** | **9** | **7** | **2** | **37.62** |

*Table 10.* Ablation results on ODE signal, sub-task ID, and planning uncertainty penalty.

### F.6 EXPERIMENT 1 & 2 - BRN GREEN & DUSTY

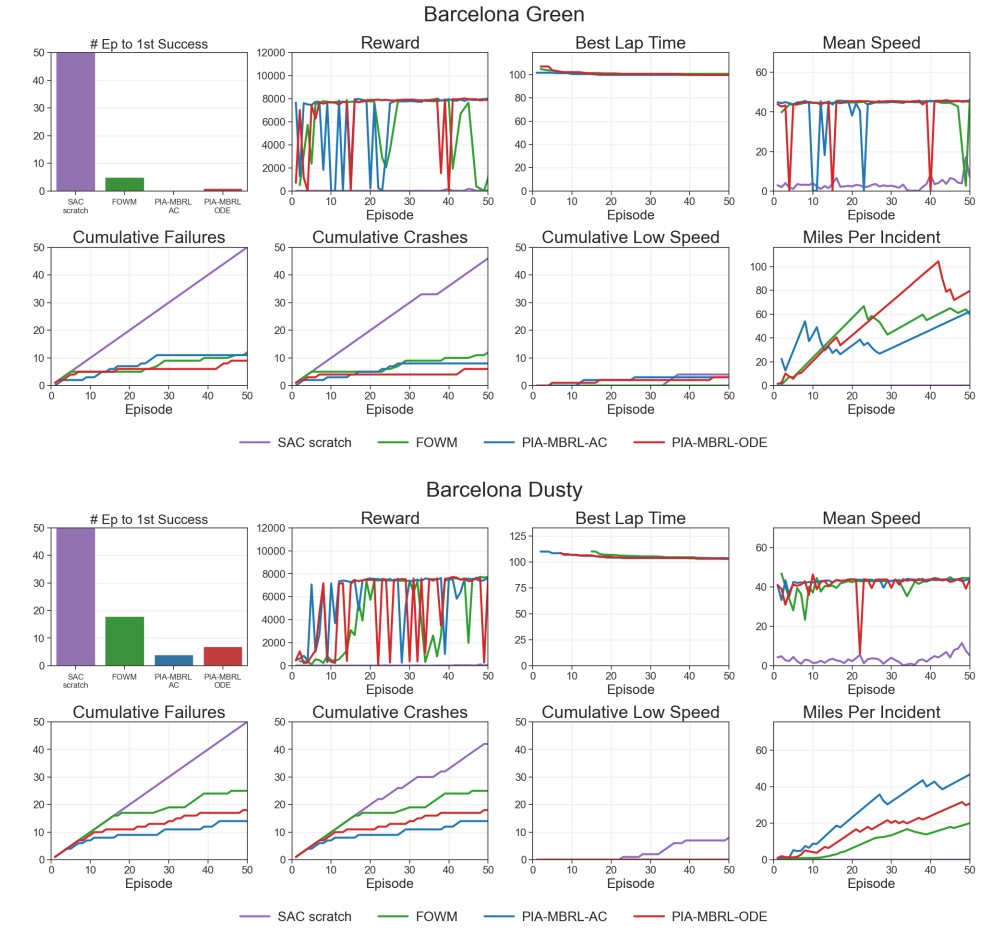

*Figure 14.* Online finetuning results: Barcelona Green Surface (top) & Dusty Surface (bottom).

*Table 11.* Online performance results across Barcelona tracks and surface conditions.

| Barcelona (Green Surface) | | | | | | | |
|---|---|---|---|---|---|---|---|
| Model | Best Lap (s) ↓ | Best Reward ↑ | # Ep to 1st Success ↓ | # Failures ↓ | # Crashes ↓ | # Low Speed ↓ | Miles per Failure ↑ |
| IQL | Fail | 453.9 | – | – | – | – | – |
| SAC scratch | Fail | 176.617 | – | 50 | 46 | 4 | 0.06 |
| FOWM | 100.636 | 7865.20 | 6 | 12 | 12 | 0 | 59.87 |
| PIA-MBRL-AC | **99.637** | 8016.66 | **1** | 11 | 8 | 3 | 62.15 |
| PIA-MBRL-ODE | 99.656 | **8026.69** | 2 | **9** | **6** | 3 | **79.42** |
| Barcelona (Dusty Surface) | | | | | | | |
| Model | Best Lap (s) ↓ | Best Reward ↑ | # Ep to 1st Success ↓ | # Failures ↓ | # Crashes ↓ | # Low Speed ↓ | Miles per Failure ↑ |
| IQL | Fail | 462.8 | – | – | – | – | – |
| SAC scratch | Fail | 105.008 | – | 50 | 42 | 8 | 0.02 |
| FOWM | 103.177 | 7729.42 | 19 | 25 | 25 | 0 | 20.02 |
| PIA-MBRL-AC | 103.48 | 7624.81 | **5** | **14** | **14** | 0 | **46.66** |
| PIA-MBRL-ODE | **103.195** | **7716.56** | 7 | 18 | 18 | 0 | 30.74 |

## F.7 ODE DATA SCALING

To understand how performance scales with ODE data size, we fix the Assetto Corsa data and progressively increase the proportion of sub-sampled ODE simulator data (0% → 25% → 50% → 75% → 100%). As shown in Figure 15, larger ODE dataset yields lower lap times and fewer incidents, which indicates a clear positive scaling trend on ODE data scalability.

However, in the low-proportion ODE regime, we also observe a trade-off between adaptation speed (#Ep To 1st Rollout) and best-lap performance: introducing a small amount of ODE data improves the eventual lap time but delays the first successful rollout. We think this occurs because the underlying dynamics and action distributions in the ODE simulator are not perfectly aligned with Assetto Corsa - with only a small fraction of ODE data (e.g., 25%), the simulator–real environment mismatch is not sufficiently compensated by the limited simulator data coverage, so the additional ODE data becomes beneficial only after the policy is re-aligned to the AC environment through online finetuning.

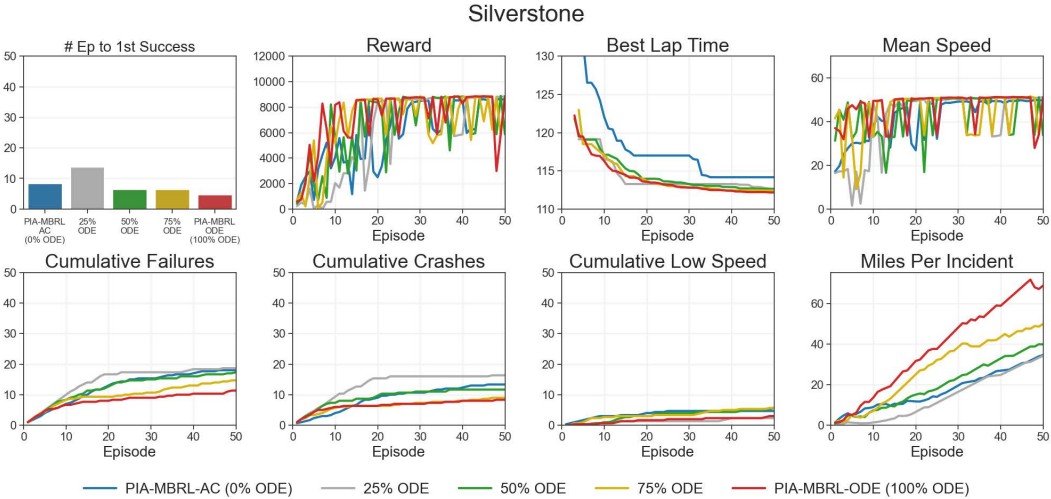

*Figure 15.* ODE data scalability results on Silverstone, with fixed Assetto Corsa dataset and different proportion of sub-sampled ODE dataset.

| Scaling ODE Dataset Size (AC Data Fixed) | | | | | | | |
|---|---|---|---|---|---|---|---|
| Model | Best Lap (s) ↓ | Best Reward ↑ | # Ep to 1st Success ↓ | # Failures ↓ | # Crashes ↓ | # Low Speed ↓ | Miles per Failure ↑ |
| 0% ODE | 114.16 | 8637.48 | 8.33 | 18 | 14 | 4 | 34.62 |
| 25% ODE | 112.66 | 8857.49 | 14.67 | 19 | 16 | 2 | 34.27 |
| 50% ODE | 112.61 | **8876.22** | 7.33 | 17 | 11 | 6 | 39.86 |
| 75% ODE | 112.42 | 8835.32 | 7.33 | 14 | 9 | 6 | 49.91 |
| 100% ODE | **112.17** | 8852.00 | **4.67** | **11** | **8** | **3** | **68.80** |

*Table 12.* ODE data scalability results on Silverstone, with fixed Assetto Corsa dataset and different proportion of sub-sampled ODE dataset.

### F.8 SAC WITH ODE DATA AUGMENTATION

In our main experiment, SAC algorithm is trained from scratch without ODE data augmentation, with Update To Data Ratio = 1.0. To further investigate how ODE data can be incorporated while mitigating bias toward the simulator, we additionally evaluate a lightweight variant, denoted **SAC+ODE**. In this setting, we first perform a short offline warm-start stage consisting of 50,000 steps gradient updates on ODE data, and then switch to the standard online SAC training procedure in Assetto Corsa interactions. Importantly, no simulator data is ever mixed into the online replay buffer; the pretraining phase merely provides an initialization prior based on ODE data.

As illustrated in Figure 17 and Table 13, the SAC+ODE variant demonstrates faster adaptation and improved performance in best lap time, number of cumulative failures and miles per incident compared to SAC trained from scratch. The algorithm benefits from pretraining on ODE data, and leads to earlier successful rollouts and stronger overall task performance. Although a sim-to-real gap exists between the physics-based ODE model and the real environment, pretraining on scalable ODE data still provides a useful initialization without changing the underlying SAC training procedure.

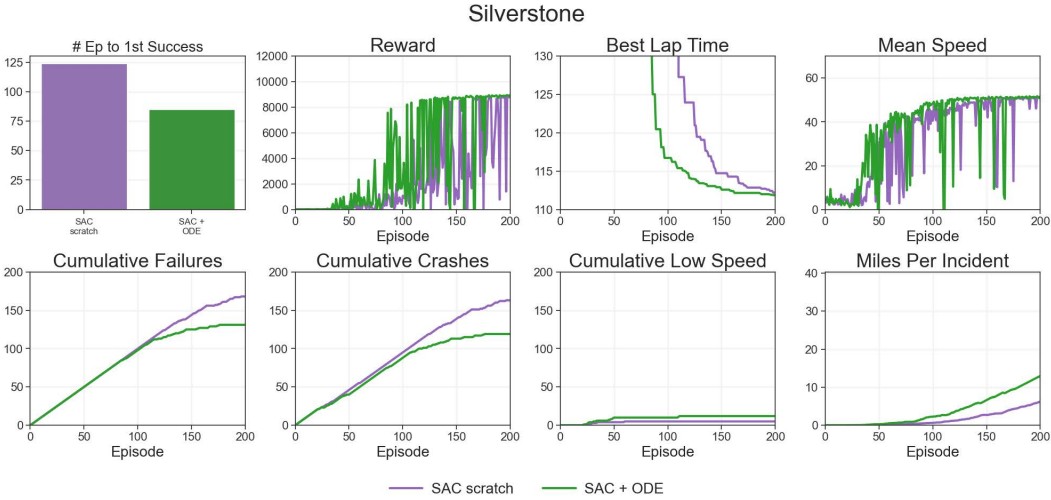

*Figure 16.* Comparison between SAC trained from scratch and SAC+ODE, where SAC+ODE receives a 50k-step offline warm-start on ODE-generated rollouts before standard online finetuning. ODE pretraining accelerates early adaptation and reduces incidents, while final performance remains governed by high-fidelity Assetto Corsa data.

| SAC Scratch vs. SAC + ODE | | | | | | | |
|---|---|---|---|---|---|---|---|
| Model | Best Lap (s) ↓ | Best Reward ↑ | # Ep to 1st Success ↓ | # Failures ↓ | # Crashes ↓ | # Low Speed ↓ | Miles per Failure ↑ |
| SAC scratch | 112.20 | 8849.34 | 125 | 168 | 163 | **5** | 6.25 |
| SAC + ODE | **111.84** | **8984.49** | **86** | **131** | **119** | 12 | **12.98** |

*Table 13.* Comparison between SAC scratch and SAC + ODE. SAC + ODE includes a 50k-step offline ODE pretraining stage, followed by standard online SAC (UTD = 1) using only real Assetto Corsa interaction.

## F.9   SIMPLIFIED ODE

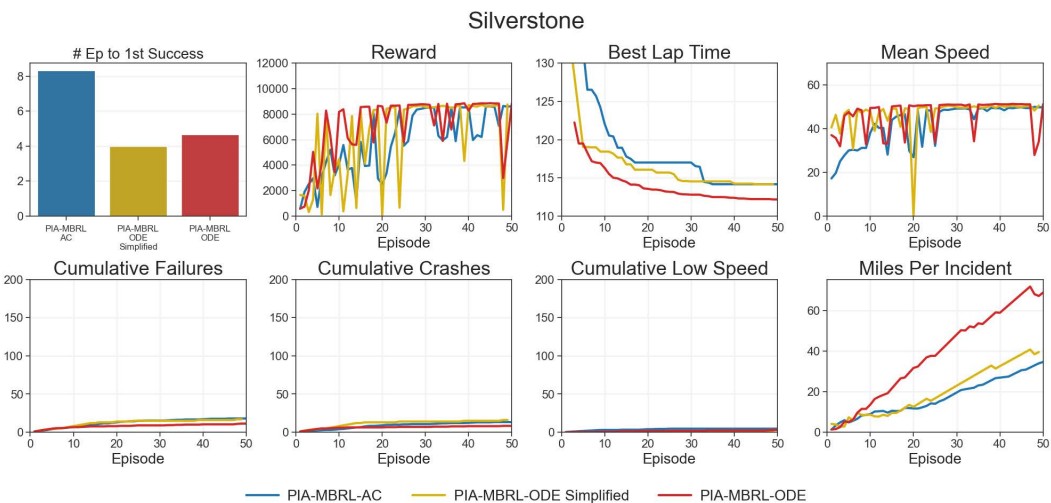

*Figure 17.* Comparison of PIA-MBRL variants with simplified ODE.

Figure 17 illustrates the effect of ODE model quality by comparing the full model with a simplified version that removes aerodynamic loads, load transfer, Pacejka nonlinearities, and friction-ellipse coupling. The simplified ODE results fall between "no ODE" and the full ODE, matching the trend observed in the geometry/dynamics ablation (Appendix F.4). Together, these results indicate a clear relationship between model fidelity and performance: simple models already provide useful structural priors, and higher-fidelity models offer additional improvements. This supports the view that the ODE does not need to match the target domain exactly, it only needs to encode the core structure of the dynamics.

| PIA-MBRL Variants on ODE Environment | | | | | | | |
|---|---|---|---|---|---|---|---|
| Model | Best Lap (s) ↓ | Best Reward ↑ | # Ep to 1st Success ↓ | # Failures ↓ | # Crashes ↓ | # Low Speed ↓ | Miles per Failure ↑ |
| PIA-MBRL-AC | 114.16 | 8637.48 | 9.33 | 18 | 13.33 | 4.67 | 34.62 |
| Simplified ODE | 114.16 | 8753.75 | 5 | 17 | 16 | 1 | 39.51 |
| PIA-MBRL-ODE | 112.17 | 8852.00 | 5.67 | 11.33 | 8.33 | 3 | 68.80 |

*Table 14.* Comparison of PIA-MBRL variants with simplified ODE.

## G   METHOD DETAILS

In this section, we provide implementation details and hyperparameters for all methods considered for our experiments.

### G.1 SAC

We base our SAC (Haarnoja et al., 2018) code on the implementation from `https://github.com/toshikwa/discor.pytorch`, making minor modifications and tuning hyperparameters. We used the hyperparameters listed in Table 15.

*Table 15.* **SAC hyperparameters.** We used the same hyperparameters across all tracks and cars.

| Hyperparameter | Value |
|---|---|
| Batch size | 128 |
| Memory size | 10,000,000 |
| Offline buffer size | 10,000,000 |
| Update interval | 1 |
| Start steps | 2000 |
| Evaluation interval | 100,000 |
| Number of evaluation episodes | 1 |
| Checkpoint frequency | 200,000 |
| Discount factor ($\gamma$) | 0.992 |
| N-step | 3 |
| Policy learning rate | 0.0003 |
| Q-function learning rate | 0.0003 |
| Entropy learning rate | 0.0003 |
| Policy hidden units | [256, 256, 256] |
| Q-function hidden units | [256, 256, 256] |
| Target update coefficient | 0.005 |
| Number of Q functions | 2 |

### G.2 IQL

We use the official implementation from Kostrikov et al. (2021) available at `https://github.com/ikostrikov/implicit_q_learning`, with minor modifications and hyperparameter tuning. This implementation uses JAX, which lacks CUDA support on Windows. Therefore, we train and evaluate the model on a Linux system while connected via Ethernet to a Windows machine running Assetto Corsa. All evaluations are conducted in the offline setting (no fine-tuning), and we assess performance every 50,000 training steps.

**Hyperparameters.** We follow the original paper's hyperparameters, tuning only the temperature and expectile ($\tau$) parameters as listed in Table 16, and increasing the buffer capacity to 50 M to accommodate all offline data. Custom parameters are highlighted in bold.

*Table 16.* **IQL Hyperparameters.** These values are consistent across all tracks and cars.

| Hyperparameter | Value |
|---|---|
| Evaluation interval | 50,000 steps |
| Batch size | 256 |
| Number of pretraining steps | $3 \times 10^6$ |
| Replay buffer size | **50,000,000** |
| Actor learning rate | $3 \times 10^{-4}$ |
| Value learning rate | $3 \times 10^{-4}$ |
| Critic learning rate | $3 \times 10^{-4}$ |
| Discount factor | **0.992** |
| Hidden layer dimensions | (256, 256, 256) |
| Temperature | **1** |
| Expectile ($\tau$) | **0.6** |

*Table 17.* **FOWM hyperparameters.** Values are consistent across all tracks and cars.

| Hyperparameter | Value |
|---|---|
| Expectile ($\tau$) | 0.6 |
| AWR temperature ($\beta$) | 1.0 |
| Uncertainty coefficient ($\lambda$) | 1 |
| $Q$ ensemble size | 5 |
| | |
| Batch size | 1024 |
| Learning rate | $3 \times 10^{-4}$ |
| Optimizer | Adam ($\beta_1 = 0.9, \beta_2 = 0.999$) |
| Discount factor | 0.992 |
| Action repeat | 1 |
| Value loss coefficient | 0.1 |
| Reward loss coefficient | 0.5 |
| Latent dynamics loss coefficient | 20 |
| Temporal coefficient ($\lambda$) | 0.5 |
| Target network update frequency | 2 |
| Polyak averaging ($\tau$) | 0.99 |
| | |
| MLP hidden size | 512 |
| Latent state dimension | 50 |
| | |
| Population size | 512 |
| Elite fraction | 50 |
| Policy fraction | 0.1 |
| Planning iterations | 1 |
| Planning horizon | 5 |
| Planning temperature | 0.5 |
| Planning momentum coefficient | 0.1 |

### G.3 FOWM

We adopt the official implementation from Feng et al. (2023) available at `https://github.com/yunhaif/fowm` with minor modifications and hyperparameter tunings.

**Hyperparameters.** We use the same hyperparameters listed in Table 17. Our adapter parameters are in bold. Specifically we set the expectile ($\tau$) to be 0.6, discount factor to be 0.992 due to our longer horizon of 15k steps per episode, and batch size to 1024 for better training performance.

### G.4 TD-MPC2

For TD-MPC2 (Hansen et al., 2023) we build on top of the publicly available JAX implementation which can be found at `https://github.com/ShaneFlandermeyer/tdmpc2-jax`. Additionally, we collected experiences and only updated the model parameters at the end of each episode. Due to hardware limitations, without this adjustment, we found that the model failed to train properly when the car was moving at higher speeds.

**Hyperparameters.** We use the same hyperparameters which are listed in Table 18. Our adapter parameters are in bold. Specifically, we set the discount factor $\gamma$ to a fixed value of 0.992, as the heuristic used in the original TD-MPC2 paper was not applicable to our tasks due to our longer horizon of 15k steps per episode. We increase the buffer capacity to 50 M to accommodate all offline data and raise the batch size to 1024 to enhance training performance. We keep the default configuration of 5 million parameters.

*Table 18.* **TD-MPC2 hyperparameters.** We use the same hyperparameters across all tracks and cars.

| Hyperparameter | Value |
|---|---|
| **Planning** | |
| Horizon ($H$) | 3 |
| Iterations | 6 |
| Population size | 512 |
| Policy prior samples | 24 |
| Number of elites | 64 |
| Minimum std. | 0.05 |
| Maximum std. | 2 |
| Temperature | 0.5 |
| Momentum | No |
| Uncertainty coef. | 0.005 |
| | |
| **Policy prior** | |
| Log std. min. | $-10$ |
| Log std. max. | 2 |
| | |
| **Replay buffer** | |
| Capacity | **50,000,000** |
| Sampling | Uniform |
| | |
| **Architecture (5M)** | |
| Encoder dim | 256 |
| MLP dim | 512 |
| Latent state dim | 512 |
| Activation | LayerNorm + Mish |
| $Q$-function dropout rate | 1% |
| Number of $Q$-functions | 5 |
| Number of reward/value bins | 101 |
| SimNorm dim ($V$) | 8 |
| SimNorm temperature ($\tau$) | 1 |
| | |
| **Optimization** | |
| Update-to-data ratio | 1 |
| Batch size | **1024** |
| Joint-embedding coef. | 10 |
| Reward prediction coef. | 0.1 |
| Value prediction coef. | 0.1 |
| Temporal coef. ($\lambda$) | 0.5 |
| Policy prior entropy coef. | $3 \times 10^{-4}$ |
| Policy prior loss norm. | Moving $(5\%, 95\%)$ percentiles |
| Optimizer | Adam |
| Learning rate | $3 \times 10^{-4}$ |
| Encoder learning rate | $3 \times 10^{-4}$ |
| Gradient clip norm | 20 |
| Discount factor | **0.992** |

