# OpenReview forum: "Leveraging Physics-Based Models for Rapid Adaptation in Reinforcement Learning"
_ICLR.cc/2026/Conference — Submitted to ICLR 2026_

### Official Review · Reviewer_wjoZ · 2025-10-24

**Soundness:** 3
**Presentation:** 2
**Contribution:** 2
**Rating:** 2
**Confidence:** 4

**Summary:**

The paper proposes a model-based framework for offline-to-online learning that leverages synthetic data that is generated from numerical solutions to dynamics ODEs (i.e. first-principles models).

The main contributions of the paper are extensive evaluation of the proposed method in the “Assetto Corsa” racing simulator. The authors also provide a faster extension of the simulator.

**Strengths:**

* The paper is easy to follow.
* The empirical results are convincing -- the method indeed works well in the Assetto Corsa Gym

**Weaknesses:**

Novelty: The method appears to have been proposed in several previous works. The primary contribution here lies in leveraging ODEs as a data augmentation technique, which differentiates this study from earlier approaches.

Specificity: While the empirical results are impressive within the race-driving context, the experiments are limited to this specific domain. Broader validation across more applications would strengthen the generality of the findings.

Assumptions: The approach assumes that the given ODEs accurately capture real-world dynamics. However, in practice, the sim-to-real gap remains a fundamental challenge when relying on first-principles models. Although the experiments provide evidence that this challenge can be mitigated, a more explicit discussion of this assumption and its implications would strengthen the paper.

A minor weakness is the repeated mention of safety concerns, which are indeed critical in race driving, without explicitly addressing this challenge.

In a broader sense, in many real-world scenarios, writing down the ODEs that govern the dynamics is too hard.
A related and potentially more impactful direction would be to augment data from physics engines that solve the underlying dynamics. A key research question then arises: how can one augment simulation-generated data intelligently, without introducing bias toward the simulated environment and thereby amplifying the sim-to-real gap?

**Questions:**

When you train SAC from scratch, do you augment it with the ODE data? What Update to Data ratio do you choose for it?

---

> ### Author Response · Authors · 2025-11-27
>
> We thank the reviewer for their valuable feedback. We address your comments in the following.
>
> **Q: Novelty of using ODEs as data augmentation**
> **A:** We would like to clarify that, to the best of our knowledge, using lightweight analytic ODE models specifically as offline augmentation for model-based RL has not been explored in prior work.
>
> **Q: Domain specificity and generality**
> **A:** We agree that broader validation is desirable. The racing domain is intentionally chosen because it is extremely challenging (fast dynamics and large distribution shifts across tracks). However, we highlight two points:
> Our method is domain-agnostic by construction: and requires only a low-cost analytic model. This applies directly to drones, quadrupeds, manipulators, etc.
> In Appendix F.8 we added experiments showing that our approach is beneficial to other RL methods beyond TD-MPC2. In particular, SAC learns faster and achieves better stability than SAC-scratch. However, the best results are consistently obtained with MBRL, which is designed for generalization and online adaptation. This matches our motivation: physics-informed augmentation provides a strong prior, but MBRL makes the best use of that prior in a multi-track, multi-surface setting.
>
> While extending to new domains is exciting future work, such expansion requires building additional simulators or physics models that are outside the scope of this submission. We will clarify this in the revised version.
>
> **Q: Assumptions about ODE accuracy & sim-to-real gap**
> **A:** We agree that analytic models do not perfectly capture real-world dynamics. This is precisely why our approach combines physics-based augmentation with online adaptation:
> The ODE is used only to bootstrap the world model with structured, physically coherent rollouts.
> The agent is then refined with a small amount of high-fidelity AC interaction, which corrects model inaccuracies.
> This mirrors standard practice in real robotics, where simplified models (for drones, quadrupeds, manipulators) are used to pretrain policies and then deployed zero shot.
>
> To support this point, we added an additional experiment: A simplified-ODE ablation (App. F.8), where we remove major modeling components. Even this simple model provides meaningful gains during pretraining, while the full ODE yields stronger robustness. This confirms that the analytic model does not need to be exact; it just needs to provide structured, physically consistent rollouts that improve initialization, with online adaptation and uncertainty-aware planning closing the remaining gap. We will further clarify this in the camera-ready version.
>
> **Q: Safety considerations**
> **A:** In our setup, safety concerns primarily relate to: avoiding catastrophic planner rollouts during early online episodes, and reducing failures (incidents) in high-speed corners.
> Our method explicitly addresses this through:
>
> - uncertainty-penalized planning, which prevents the agent from committing to model predictions far from the training distribution, and
> - reduced real-interaction requirements, meaning fewer unsafe exploratory episodes.
>     We will clarify this connection in the camera-ready version.
>
> **Q: In a broader sense, in many real-world scenarios, writing down the ODEs that govern the dynamics is too hard**
> **A:** The reviewer suggests using physics-engine rollouts (e.g., MuJoCo, Isaac) as an interesting direction.
> We agree that this is valuable, but note that most physics engines are themselves built on hand-coded ODEs / rigid-body dynamics.
>
> **Q: potentially more impactful direction would be to augment data from physics engines that solve the underlying dynamics.**
> **A:** Augmenting data from physics engines is indeed a promising direction, especially for domains where hand-written ODEs are difficult to construct. Exploring physics-engine–based augmentation is interesting future work.
>
> **Q: A key research question then arises: how can one augment simulation-generated data intelligently, without introducing bias toward the simulated environment and thereby amplifying the sim-to-real gap?**
> **A:** Our current approach already addresses this concern: online adaptation and uncertainty-aware planning prevent the model from relying too heavily on synthetic data when it encounters high-fidelity trajectories. These mechanisms naturally weight down synthetic rollouts that do not transfer well. However, we agree that there is potential for improvement in this direction.
>
> **Q: SAC-from-scratch question**
> **A:** We added a new experiment in Appendix F.8 addressing this directly. We ran SAC from scratch with and without ODE augmentation.
> SAC without ODE learns significantly slower. SAC with ODE augmentation uses ODE transitions and a 1:1 update-to-data ratio during the early stages.
>
> Please do not hesitate to let us know if you have any additional comments.
>
> ---

---

> > ### Comment · Reviewer_wjoZ · 2025-11-28
> >
> > I would like to thank again the authors for their thorough response.
> >
> > > Q: In a broader sense, in many real-world scenarios, writing down the ODEs that govern the dynamics is too hard
> > A: The reviewer suggests using physics-engine rollouts (e.g., MuJoCo, Isaac) as an interesting direction.
> > We agree that this is valuable, but note that most physics engines are themselves built on hand-coded ODEs / rigid-body dynamics.
> >
> > Thank you for your response. One could argue that such simulators are built on hand-coded analytical dynamics equations, which is a valid statement. However, these equations are not derived manually nor analytically: they are usually transpiled from xml specifications that sometime include highly complex systems, which---practically speaking---cannot be derived manually. This is precisely the point that I wanted to make, that relying on more complex simulators, rather than on analytical equations would be a more impactful direction to study. This is because most (interesting) dynamical systems are hard to derive without making major simplifications. In other words, instead of limiting the scope of research only to simulators that are derived analytically, a framework that studies simulators in their full generality could be more relevant.
> >
> > > Q: Novelty of using ODEs as data augmentation
> > A: We would like to clarify that, to the best of our knowledge, using lightweight analytic ODE models specifically as offline augmentation for model-based RL has not been explored in prior work.
> >
> > I agree with this statement. At the same time I couldn't identify major conceptual innovations in the proposed method. For instance, uncertainty-penalized planning has already been studied before [1].
> >
> > > Q: SAC-from-scratch question
> > A: We added a new experiment in Appendix F.8 addressing this directly. We ran SAC from scratch with and without ODE augmentation.
> > SAC without ODE learns significantly slower. SAC with ODE augmentation uses ODE transitions and a 1:1 update-to-data ratio during the early stages.
> >
> > Please note that is has been shown several times that by simply increasing the UTD (e.g. 1:4) one can already significantly improve SAC's sample efficiency [2]. If I understood the experiment correctly, this could be a more appropriate/timely comparison.
> >
> > I would like to add a few positive words on this paper.
> >
> > Overall, one can see that there has been a major work on the paper:
> > * The experiments are comprehensive.
> > * The figures are neat.
> > * Text/writing is overall streamlined and easy to follow.
> >
> > That is to say that the paper itself is of good quality! It is mainly its contribution to the RL community that, in my view, could be strengthened:
> > * The methods used are (generally speaking) standard.
> > * While well-made, the empirical study is in some sense limited since it uses only one application domain and only analytical simulators. Could you show such significant improvement across benchmarks/tasks? That could be useful for many researchers.
> >
> > In its current form, I could imagine this work being better received in a more application-focused venue, where the strength of the paper---its careful engineering, clear setup, and thorough experiments on a concrete class of systems---would likely be a primary evaluation criterion.
> >
> > ---
> > [1] Yu, Tianhe, Garrett Thomas, Lantao Yu, Stefano Ermon, James Y. Zou, Sergey Levine, Chelsea Finn, and Tengyu Ma. "Mopo: Model-based offline policy optimization." Advances in Neural Information Processing Systems 33 (2020): 14129-14142.
> >
> > [2] Rybkin, Oleh, Michal Nauman, Preston Fu, Charlie Snell, Pieter Abbeel, Sergey Levine, and Aviral Kumar. "Value-based deep rl scales predictably." arXiv preprint arXiv:2502.04327 (2025).

---

### Official Review · Reviewer_mYmD · 2025-10-28

**Soundness:** 3
**Presentation:** 2
**Contribution:** 2
**Rating:** 4
**Confidence:** 4

**Summary:**

This paper presents PIA-MBRL (Physics-Informed Augmentation with Model-Based RL), a framework for rapidly adapting reinforcement learning agents to new tasks by integrating lightweight analytical models into a model-based RL (MBRL) pipeline. The key insight is to use stable ODE-based vehicle models to generate physics-grounded synthetic rollouts, which are merged with limited high-fidelity simulator data (Assetto Corsa Gym) to improve sample efficiency and generalization. The approach builds on TD-MPC2 and the offline-to-online adaptation paradigm of FOWM. Tested in diverse conditions within the autonomous car racing scenario, the method achieves faster convergence and improved task performance.

Experiments in autonomous racing show that physics-informed augmentation enables agents to adapt to unseen tracks and surface conditions (e.g., dusty, low-grip) within a few episodes. The authors release over 1000 hours of racing data and ODE rollouts, along with an enhanced version of ACGym that supports dynamic model swapping and JAX-based real-time control.

**Strengths:**

- Justified design choice: Uses analytic ODE models as reliable, long-horizon data generators for MBRL. This idea is well-suited for the racing task.

- Engineering contribution: Extends ACGym with Linux support, modular dynamics swapping, and real-time TD-MPC2 in JAX.

- Extensive eval Thorough experiments across 15 tracks, 3 surface conditions, and multiple baselines (SAC, IQL, FOWM, TD-MPC2).

**Weaknesses:**

- Limited novelty in algorithmic form: Builds directly upon existing frameworks (TD-MPC2, FOWM), with the main innovation being the physics-based data source.
- Also, the methodology is only tested on a single task in a simulation environment. Based on the limited domain demonstrated, the paper's title should also incorporate the specific domain, e.g., add "...for autonomous racing"
- Another disadvantage of using a strong model prior assumption is the unmodeled effects. ODE models use fixed parameters and do not adapt to varying surface frictions or uncertainties. Despite being more efficient in terms of learning efficiency and even final performance, I do see that this approach has rather limited expressiveness when encountering more complex models.
- Generalization claims: While racing is a strong testbed, broader applicability (e.g., robotics or UAVs) remains unverified.

Apart from the methodology section,
- Sim2real verification: An alternative way to significantly improve the paper is to demonstrate that the approach can perform some sim2real verification to demonstrate the approach's effectiveness (not even necessarily for car racing at all, it can be even simpler tasks). This will strongly support the model learned, which can handle real-world scenarios.

**Questions:**

See the weaknesses above. I will be happy to see the paper further improved.

---

> ### Author Response · Authors · 2025-11-27
>
> We thank the reviewer for their valuable feedback. We address your comments in the following.
>
> **Q: Limited novelty in algorithmic form**
> **A:** While our framework builds upon existing components (offline→online RL, TD-MPC2, FOWM), our contribution is how these pieces are combined with physics-based synthetic data. To the best of our knowledge, using lightweight analytic ODEs as large-scale, long-horizon data sources for MBRL adaptation in high-speed racing has not been previously explored. As shown in our ablations (Sec. 4.2–4.3, App. F), the augmentation signal provided by the ODE is essential for both sample efficiency and safety, and the gains do not appear when removing the physics prior.
>
> **Q: Only tested in a single domain; title should include ‘for autonomous racing**
> **A:** We can add “for autonomous racing” in the camera-ready if the AC remains consistent across reviewers, but we note that racing is not a toy domain; it is a fast, highly nonlinear system with high-frequency dynamics, actuator limits, load transfer, combined slip, and aerodynamics. Our method is domain-agnostic by design (it only requires an ODE prior). Given the evaluation across 15 tracks × 3 surfaces, we believe the current title reflects the method’s generality while demonstrating it in a strong testbed. If AC reviewers converge on asking for domain specificity, we will update the title.
>
> **Q: Strong model prior; limited expressiveness; unmodeled effects**
> **A:** We agree that analytic ODEs are approximate and cannot capture all effects (surface friction variation, transient tire behavior, load changes). This is exactly why our method includes online adaptation and uncertainty-aware planning. The ODE acts as a prior, not as a ground-truth simulator.
>
> Two of our ablations make this explicit:
> a. Geometry vs. dynamics vs. both (App. F.4): even partial physics helps, but higher fidelity improves robustness. b. Simplified ODE (App. F.9): even a very reduced model provides meaningful gains, though full ODE performs best.
>
> This supports the claim that the ODE need not be perfect, it only needs to provide consistent structure that the MBRL agent refines during online interaction. Racing is already a complex system with strong unmodeled effects, and the fact that our approach improves performance and sample efficiency despite these mismatches strengthens the contribution.
>
> **Q: Generalization claims beyond racing remain unverified**
> **A:** We agree this is an important direction. Our method is conceptually domain-agnostic, but evaluating additional domains requires substantial additional simulators and priors. We list UAVs, manipulation, and quadrupeds as future work and such extensions are beyond the scope of this paper and would each require their own full experimental section.
>
> **Q: Sim2real verification would strengthen the paper**
> **A:** We completely agree that sim2real evaluation would further reinforce the claims.
> Assetto Corsa is already a high-fidelity, physics-based driving simulator, and many real-racing pipelines use it as a proxy for real-vehicle dynamics. Sim2real evaluation with our real vehicle is part of our roadmap, but: performing real-vehicle experiments for 15 tracks × 3 surfaces is infeasible within the timeframe of an academic submission, and sim2real robotics papers often rely on massively simplified tasks, whereas our domain is inherently high-speed and safety-critical.
>
> We will add a short discussion clarifying this.
>
> We appreciate the reviewer’s near-positive stance and their constructive suggestions.
> We have added several of the requested ablations (simplified ODE, scaling, geometry vs dynamics) and clarified scope, novelty, and limitations throughout the revised paper.
>
> Please do not hesitate to let us know if you have any additional comments.

---

### Official Review · Reviewer_uPNz · 2025-10-31

**Soundness:** 4
**Presentation:** 4
**Contribution:** 2
**Rating:** 8
**Confidence:** 4

**Summary:**

The paper proposes an approach that leverages low-fidelity simulators to achieve more sample-efficient reinforcement learning on real systems. The algorithm first trains a policy offline using a dataset composed of both simulated and real-world data. This pretrained policy is then fine-tuned through online learning on the real system. The authors demonstrate that incorporating simulated data substantially improves key RL metrics such as safety and sample efficiency. They evaluate their method using the Assetto Corsa simulator as a proxy for the real system and employ an advanced bicycle model (including aerodynamics, drivetrain, and Pacejka tire dynamics) as the low-fidelity simulator. Finally, they conduct several ablation studies and experiments to validate the design choices and highlight the benefits of their approach.

**Strengths:**

The paper is clearly written and easy to follow. Each design decision is thoroughly ablated. The authors present a comprehensive set of experiments on the Assetto Corsa simulator. The simplicity of their method is another strong point.

**Weaknesses:**

It is not clear how similar the dynamics of the Assetto Corsa simulator are to those of their bicycle model.
In the result tables, the authors report only mean values; however, many of the numbers are quite close, and it would be important to include measures of variability (e.g., standard deviations or confidence intervals) to assess whether the reported improvements are statistically significant or fall within the range of uncertainty.
Several related works in the literature address similar problems and should be discussed in the related work section [1, 2, 3, 4].
Finally, evaluating the proposed method on a broader set of dynamical systems, as mentioned in the paper, would strengthen the empirical evidence for its general applicability.

[1] Rothfuss, Jonas, et al. "Bridging the sim-to-real gap with Bayesian inference." 2024 IEEE/RSJ International Conference on Intelligent Robots and Systems (IROS). IEEE, 2024.
[2] Hwangbo, Jemin, et al. "Learning agile and dynamic motor skills for legged robots." Science Robotics 4.26 (2019): eaau5872.
[3] Ha, Sehoon, and Katsu Yamane. "Reducing hardware experiments for model learning and policy optimization." 2015 IEEE International Conference on Robotics and Automation (ICRA). IEEE, 2015.
[4] Pastor, Peter, et al. "Learning task error models for manipulation." 2013 IEEE International Conference on Robotics and Automation. IEEE, 2013.

**Questions:**

Could you perform an ablation study on your simulator? It appears that you used a rather advanced bicycle model. Would a simpler model (e.g., one that does not model tire slippage or Pacejka tire dynamics) lead to similar learning performance?

In Figure 6, there seems to be a trade-off between the proportion of simulator data and real data used in the offline phase. Could you, for example, fix the amount of Assetto Corsa (AC) data and plot the performance as the amount of ODE data increases—that is, show how performance scales with a growing simulator dataset while keeping the real data constant?

---

> ### Author Response · Authors · 2025-11-27
>
> We thank the reviewer for their valuable feedback. We address your comments in the following.
>
> **Q: Ablation with a simpler ODE (no slip, no Pacejka)**
> **A:** We added in Appendix F.9 an ablation comparing the full ODE model with a simplified version that removes aerodynamic loads, load transfer, Pacejka nonlinearities, and friction‐ellipse coupling. The simplified ODE results fall between “no ODE” and the full ODE, matching the trend observed in the geometry/dynamics ablation (Appendix F.4). Together, these results indicate a clear relationship between model fidelity and performance: simple models already provide useful structural priors, and higher-fidelity models offer additional improvements. This supports the view that the ODE does not need to match the target domain exactly.
>
> **Q: Scaling experiment: fixed AC data, increasing ODE data**
> **A:** As suggested, we added an experiment where we fix the AC dataset size and vary the amount of ODE data (Appendix F.7).
>
> We observe clear gains from moderate amounts of ODE data while larger ODE dataset yields lower lap times and fewer incidents, which indicates a clear positive scaling trend on ODE data scalability.
>
> **Q: Variability / confidence intervals**
> **A:** We appreciate this point. Due to compute constraints, many of our large-scale multi-track experiments were run with a single rollout per condition, which prevented meaningful variance reporting for those cases.
>
> To partially mitigate this limitation: Our evaluation spans 15 tracks × 3 surfaces, which provides cross-task consistency as a form of robustness check. The performance improvements are consistently across tasks, suggesting they are not due to noise from a single run.
>
> Lap time and incident counts are also stable indicators in this domain, since each episode starts from different positions and scenarios on the track.
>
> **Q: Missing related work**
> **A:** Thank you for pointing out these works. We will incorporate them into the camera-ready.
>
> **Q: Broader validation beyond racing**
> **A:** We agree that exploring additional dynamical systems would further support generality.
>
> Our method is domain-agnostic by design, but extending it to new systems (e.g., drones, manipulators) requires additional high-fidelity simulators or analytic models, which is beyond the scope of this paper. We view this as a promising direction for future work, including eventual sim-to-real experiments with the real vehicle.
>
> We thank the reviewer again for the positive evaluation and helpful suggestions. The requested ablations have been added, and the remaining points are acknowledged and will be addressed in the camera-ready version.
>
> Please do not hesitate to let us know if you have any additional comments.

---

### Official Review · Reviewer_LAm6 · 2025-10-31

**Soundness:** 1
**Presentation:** 1
**Contribution:** 1
**Rating:** 2
**Confidence:** 4

**Summary:**

The paper proposes PIA-MBRL (Physics-Informed Augmentation for Model-Based Reinforcement Learning), a framework for efficient MBRL with physics informed models. The approach brings together many techniques from the MBRL literature. The paper demonstrates the approach on primarily in a high-fidelity driving simulator, and demonstrates that a simplified ODE model can be used for effective data augmentation in this setting.

**Strengths:**

- The paper evaluates methods across a wide range of different tracks
- The paper provides an open source simulation environment for future work

**Weaknesses:**

$\textbf{Clarity}$: After two close readings of the paper, I do not understand the details of the method or the logic behind the approach. The paper appears to be bringing together a number of different techniques, such as offline to online fine-tuning, online planning with a model (via TDMPC-2), Dyna-style data augmentation and physics informed models. The draft mentions that each of these techniques are used at a high level, but does not explicitly lay out exactly how these pieces fit together.


$\textbf{Motivation}$: The method appears to be a “soup” of many existing techniques, and seems overly complex to implement. The paper does little to explain why all of these different techniques are needed together. There are no clear technical take aways from the paper that came through during my reading.

**Questions:**

- Can the authors clarify how each of the components of the framework fit together?


- Why do we need all of these pieces together?


- Why not use a simpler approach, and simply do MPC with a learned physics-based model like in numerous prior works?

---

> ### Author Response · Authors · 2025-11-27
>
> ### **Q: Can the authors clarify how each of the components of the framework fit together?**
>
> ### **Q: The method appears to be a “soup” of many existing techniques, and seems overly complex to implement.**
>
> **A:** We respectfully disagree with the characterization that the method is a “soup.” Combining several well-understood components is common in impactful RL work—methods such as **Rainbow** and **Dreamer** also integrate multiple existing ideas into a single pipeline. Their contributions lie in _how_ these pieces are assembled and validated at scale.
>
> We are the first to show that a carefully designed combination of **physics-based priors**, **offline learning**, and **online adaptation** provides substantial practical benefits in a challenging, high-speed control domain.
>
> **PIA-MBRL** is organized around a single central idea: **Reduce real-interaction requirements for adapting racing policies by using lightweight physics models as a scalable, reliable source of synthetic experience.**
>
> Each component directly supports this goal:
> - **Physics-informed augmentation:** provides structured synthetic data
> - **Offline pretraining:** leverages this data before deployment
> - **Online adaptation:** transfers to new tracks and surfaces
> - **Uncertainty regularization + data mixing:** stabilizes early adaptation
>
> Removing any component breaks the pipeline.
> The framework is a **three-stage pipeline** (Algorithm 1):
> 1. Build offline buffer with AC + ODE data.
> 2. Offline pretrain a world model and policy (with task IDs for track/source).
> 3. Online adapt using a small amount of real interaction and uncertainty-aware planning.
>
> We will revise Section 3 to present this flow more clearly in the camera-ready version.
>
> ---
>
> ### **Q: Why do we need all these pieces together?**
>
> **A:** We agree that this must be demonstrated, and we provide ablations and comparisons isolating each component (Sec. 4.2–4.3, App. F.4–F.8):
> - **ODE augmentation vs. no ODE:**
>     PIA-MBRL-ODE consistently improves lap time and reduces incidents relative to PIA-MBRL-AC (no ODE), especially on difficult tracks and low-grip surfaces (Tables 1, 2, and 7).
> - **Geometry vs. dynamics vs. both:**
>     The ODE-signal ablation (App. F.4, Table 10) shows:
>     – _geometry-only_ improves safety but not lap time,
>     – _dynamics-only_ underperforms on both,
>     – _geometry + dynamics_ (ours) yields the strongest performance and robustness.
> - **Planner regularization and task/context IDs:**
>     Removing either degrades stability and generalization (App. F.4, Table 10), increasing failures and worsening lap times.
> - **Backbone choice vs. SAC / IQL / FOWM:**
>     TD-MPC2 with our offline:online scheme outperforms SAC-scratch, IQL, and FOWM across all tracks and surfaces (Tables 1–2, 7).
>     SAC+ODE (App. F.8) shows small gains from ODE seeding but still lags significantly behind our approach.
>
> These results show that PIA-MBRL is **not** an arbitrary mixture of techniques. Each component contributes measurable improvements, and removing any one consistently degrades performance.
>
> ---
>
> ### **Q: Why not simply use MPC with a physics-based model?**
> **A:** We agree that MPC with an analytical model is an important baseline. In fact, the **ACGym** paper, on which our ODE bicycle model is based, already includes such an MPC baseline. That work reveals two key limitations motivating PIA-MBRL:
>
> **(a) MPC was significantly slower and less performant than learned methods.**
> ACGym reports that even a hand-tuned MPC using an explicit racing line performs worse than learned policies on lap time and robustness. It also requires heavy per-track tuning, making it difficult to scale across diverse tracks and surfaces. Exactly our target setting.
>
> **(b) Learned policies outperform MPC but require impractical amounts of real interaction.**
> This is the core gap we aim to close:
>
> - **MPC:** data-free but brittle and slow
> - **Learned policies:** strong but too data-hungry for real racing or high-fidelity sim
>
> **(c) Our ablations confirm that “ODE-only control” is insufficient.**
> In Fig. 3 (“Synthetic only”), pretraining with ODE-only data yields poor lap times and high failure rates. This mirrors the ACGym MPC result: an analytical model alone cannot support high-speed racing in complex environments.
>
> The performance gains arise from **MBRL + offline augmentation + online adaptation**, not from using the ODE as a standalone controller.
> Together, these results justify why a pure MPC-with-physics baseline is not our focus:
>
> - ACGym already shows its limitations
> - Our ODE-only ablation reproduces those limitations
> - Our contribution lies in using the physics model as a scalable _data source_, not as the controller
> ---
>
> Please do not hesitate to let us know if you have any additional comments.

---

### Author Response · Authors · 2025-11-27

We thank all reviewers for their thoughtful comments. We have revised our manuscript based on your feedback – the list of changes are available below. We have also responded to your individual comments.

**Summary of the revision:**

Experiments added in this revision (included in the uploaded PDF):
- ODE fidelity ablation (Appendix F.9) using a simplified model
- Scaling experiment with fixed AC data and increasing ODE data (Appendix F.7).
- SAC+ODE experiment (Appendix F.8)

Several recurring concerns were brought up across reviewers, and we have addressed them in our responses:
- ODE accuracy & sim-to-real gap / strong model prior: reviewers questioned the reliance on an analytic ODE. We clarified that the ODE is used strictly as a prior, not an exact simulator, and that online adaptation + uncertainty-aware planning corrects unmodeled effects. The new simplified-ODE ablation supports this defense by showing that even coarse models help.
- Domain specificity & generality: reviewers noted the evaluation is in racing only. We clarified that the method is domain-agnostic by design, but extending it to UAVs, manipulators, or quadrupeds requires additional simulators/ODEs and is thus beyond the scope of this work. We will position these directions explicitly as future work.

Again, we thank the reviewers for their constructive feedback. We believe that all comments have been addressed in this revision, but are happy to address any further comments from reviewers.

Best,
Authors of “Leveraging Physics-Based Models for Rapid Adaptation in Reinforcement Learning”

---

### Meta-Review · Area_Chair_xenX · 2026-01-02

**Summary:**

Most reviewers raise concerns about novelty. The paper is a well written system paper, which applies physics based model to simulated racing. On the algorithmic side, however, the novelty is limited to using "physics based models" to a framework that is largely based on existing algorithms like TD-MPC2 and FOWM. While I agree with the authors that this idea can be non-trivial from a system aspect, I also agree with the reviewers that it brings limited innovation from a machine learning or reinforcement learning perspective. For the latter, I think the authors need to demonstrate the same idea can be applicable to a broader context. As reviewer wjoZ  suggested, I also think that this paper would have a larger impact in a more application-focused venue.

**Reviewer Concerns:**

The reviewers' main concern is the algorithmic novelty. The criticism is tied to the nature of the paper. I do not think the rebuttal would change this opinion.

**Reviewer Scores:**

I think they would likely remain their scores.

---

### Decision · Program_Chairs · 2026-01-26

Reject